# BAYESIAN META SAMPLING FOR FAST UNCERTAINTY ADAPTION

**Zhenyi Wang** [1]**, Yang Zhao** [1]**, Ping Yu** [1]**, Ruiyi Zhang** [2]**, Changyou Chen** [1]
[1] State University of New York at Buffalo          [2] Duke University
[1] `{zhenyiwa, yzhao63, pingyu, changyou}@buffalo.edu`
[2] `ryzhang@cs.duke.edu`

## ABSTRACT

Meta learning has been making impressive progress for fast model adaptation. However, limited work has been done on learning fast uncertainty adaption for Bayesian modeling. In this paper, we propose to achieve the goal by placing meta learning on the space of probability measures, inducing the concept of meta sampling for fast uncertainty adaption. Specifically, we propose a Bayesian meta sampling framework consisting of two main components: a meta sampler and a sample adapter. The meta sampler is constructed by adopting a neural-inverse-autoregressive-flow (NIAF) structure, a variant of the recently proposed neural autoregressive flows, to efficiently generate *meta samples* to be adapted. The sample adapter then pushes meta samples to task-specific samples, based on a newly proposed and general Bayesian sampling technique, called optimal-transport Bayesian sampling. The combination of the two components allows a simple learning procedure for the meta sampler to be developed, which can be efficiently optimized via standard back-propagation. Extensive experimental results demonstrate the efficiency and effectiveness of the proposed framework, obtaining better sample quality and faster uncertainty adaption compared to related methods.

## 1 INTRODUCTION

Meta learning (Schmidhuber, 1987; Andrychowicz et al., 2016) is an important topic in modern machine learning. The goal is to learn some abstract concepts from different but related tasks, which can then be adapted and generalized to new tasks and environments that have never been encountered during training. There has been lots of research on this topic. A recent review classifies the methods as metric-based, model-based and optimization-based methods (Weng, 2018). Among these methods, learning-to-learn seeks to learn a meta optimizer that can be applied to different models, with some task-specific information such as current gradients as input (Andrychowicz et al., 2016). Model agnostic meta learning (MAML) aims to learn a *meta parameter/model* from a set of training tasks such that it can quickly adapt to models for new tasks (Finn et al., 2017). Many follow-up works have been proposed recently, including but not limited to the meta network (Munkhdalai & Yu, 2017), the meta learner (Ravi & Larochelle, 2017), the Reptile model (Nichol et al., 2018), and the lately extensions to an online setting (Finn et al., 2019), to model hierarchical relation (Yao et al., 2019) and sequential strategies (Ortega et al., 2019), and to its stable version Antoniou et al. (2019) and to some theoretical analysis (Khodak et al., 2019).

It is worth noting that all the aforementioned models are designed from an optimization perspective. Bayesian modeling, in parallel with optimization, has also been gaining increasing attention and found various applications in deep learning. Recent research has extended the above meta-learning methods to a Bayesian setting. For example, Bayesian MAML (BMAML) replaces the stochastic-gradient-descent (SGD) step with Stein variational gradient descent (SVGD) for posterior sampling (Yoon et al., 2018). Probabilistic MAML (PMAML) extends standard MAML by incorporating a parameter distribution of the adapted model trained via a variational lower bound (Finn et al., 2018). Amortized Bayesian Meta Learning extends the idea of MAML to amortized variational inference (Ravi & Beatson, 2019; Choi et al., 2019). VERSA (Gordon et al., 2019) uses an amortization network to approximate the posterior predictive distributions. Meta particle flow (Chen et al., 2019) realizes Bayes's rule based on ODE neural operator that can be trained in a meta-learning framework.

Though methodologically elegant with many interesting applications, the above methods lack the ability to uncertainty propagation/adaption, in the sense that uncertainty is either not considered (*e.g.*, in MAML) or only considered in the specific task level (*e.g.*, BMAML). Uncertainty modeling is critical when considering especially in few labeled data setting. Lacking of such modeling, the prediction uncertainty would be inaccurate. Model prediction uncertainty is considered and calibrated in Ananya Kumar (2019).

To tackle this problem, we propose to perform meta learning on the space of probability measures, *i.e.*, instead of adapting parameters to a new task, one adapts a meta distribution to new tasks. When implementing distribution adaption in algorithms where distributions are approximated by samples, our distribution-adaptation framework becomes sample-to-sample adaption. In other words, the meta parameter in standard MAML becomes meta samples in our method, where uncertainty can be well encoded. For this reason, we call our framework Bayesian meta sampling.

Specifically, we propose a mathematically elegant framework for Bayesian meta sampling based on the theory of Wasserstein gradient flows (WGF) (Ambrosio et al., 2005). Our goal is to learn a meta sampler whose samples can be fast adapted to new tasks. Our framework contains two main components: a *meta sampler* and a *sample adapter*. For the meta sampler, we adopt the state-of-the-art flow-based method to learn to transport noise samples to meta samples. Our meta sampler is parameterized by a neural inverse-autoregressive flow (NIAF), an extension of the recently developed neural autoregressive flows (NAFs) (Huang et al., 2018). The NIAF consists of a meta-sample generator and an autoregressive conditioner model, which outputs the parameters of the meta-sample generator. The NIAF takes some task-specific information (such as gradients of target distributions) and random noise as input and outputs meta samples from its generator. These meta samples are then quickly adapted to task-specific samples of target distributions by feeding them to the *sample adapter*. To ensure efficient and accurate adaptations to new task distributions, a novel optimal-transport Bayesian sampling (OT-sampling) scheme, based on Wasserstein gradient flows, is proposed as the adaptation mechanism of the sample adapter. The OT-sampling is general and can ensure samples to be adapted in a way that makes the sample density evolve to a target distribution optimally, thus endowing the property of fast uncertainty adaption. Finally, when one aims to perform specific tasks such as Bayesian classification with a task network, these samples are used to encode uncertainty into modeling. To this end, we further develop an efficient learning algorithm to optimize the task network based on variational inference. Extensive experiments are conducted to test the advantages of the proposed meta-sampling framework, ranging from synthetic-distribution to posterior-distribution adaption and to $k$-shot learning in Bayesian neural networks and reinforcement learning. Our results demonstrate a better performance of the proposed model compared to related methods.

## 2 NEURAL INVERSE-AUTOREGRESSIVE BAYESIAN META SAMPLING

Our model combines ideas from Bayesian sampling, Wasserstein gradient flows and inverse-autoregressive flows. A detailed review of these techniques is provided in Section A of the Appendix.

### 2.1 THE BASIC SETUP AND OVERALL IDEA

In meta sampling, one is given a set of related distributions, *e.g.*, posterior distributions of the weights of a set of Bayesian neural networks (BNNs), each of which is used for classification on a different but related dataset. With our notation, each of the network and the related dataset is called a task,

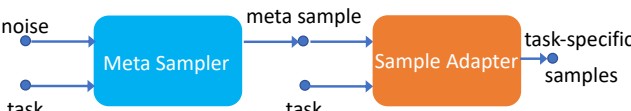

Figure 1: Overall idea of the proposed meta-sampling framework. The task is denoted with $\tau$. The two components and specific inputs will be described in details.

which is denoted as $\tau$. Meta sampling aims to learn a *meta sampler* based on a set of training tasks so that samples from the meta sampler can be fast adapted to samples for an unseen new task.

For each task, the dataset $D_\tau$ is divided into two sets $D_\tau^{tr} \triangleq \{X_\tau^{tr}, y_\tau^{tr}\}$ and $D_\tau^{val} \triangleq \{X_\tau^{val}, y_\tau^{val}\}$. Our model builds on the MAML framework Finn et al. (2017), consisting of meta initialization and task adaptation. The difference is that instead of adapting parameters as in MAML, we adapt the distribution of parameters in terms of samples. Following Grant et al. (2018), the probabilistic view

of our model can be formulated as a hierarchical Bayes model in equation 1, where $\mathbf{W}_0$ denotes the meta initialization of network parameters, $\mathbf{W}_\tau$ denotes the network parameters for task $\tau$, and adapting from $P(\mathbf{W}_0 | D_{\mathcal{T}}^{tr})$ to $P(\mathbf{W}_\tau | D_\tau^{tr})$ for a specific task $\tau$ is denoted as $P(\mathbf{W}_\tau | D_\tau^{tr}, \mathbf{W}_0)$:

$$P(y_{\mathcal{T}}^{val} | X_{\mathcal{T}}^{val}, D_{\mathcal{T}}^{tr}) = \underset{\mathbf{W}_0 \sim P(\mathbf{W}_0 | D_{\mathcal{T}}^{tr})}{\mathbb{E}} \prod_{\tau \in \mathcal{T}} \int P(y_\tau^{val} | \mathbf{W}_\tau, X_\tau^{val}) P(\mathbf{W}_\tau | D_\tau^{tr}, \mathbf{W}_0) d\mathbf{W}_\tau \ . \quad (1)$$

Our overall idea to solve the problem is to mimic a hierarchical-sampling procedure but in a much more efficient way. Specifically, we propose to decompose meta sampling into two components: a meta sampler and a sample adapter. The *meta sampler* is responsible for generating *meta samples* that characterize common statistics of different tasks, *i.e.*, approximating $P(\mathbf{W}_0 | D_{\mathcal{T}}^{tr})$ on all the training tasks $\mathcal{T}$ with meta samples; The *sample adapter* is designed for fast adaptation of meta samples to task-specific target distributions, *i.e.*, adapt the distribution from $P(\mathbf{W}_0 | D_{\mathcal{T}}^{tr})$ to $P(\mathbf{W}_\tau | D_\tau^{tr})$ in terms of samples for a task $\tau$. The meta sampler is parameterized as a conditional generator, and aggregates all local losses of different tasks to form a final loss for optimization based on optimal-transport theory. Our method allows gradients to be directly backpropagated for meta-sampler updates. The overall idea is illustrated in Figure 1.

**Comparisons with related works**  We distinguish our model with two mostly related works: the meta NNSGHMC (Gong et al., 2019) and the probabilistic MAML (PMAML) (Finn et al., 2018). The main differences lie in two aspects: meta representation and model architecture. In terms of meta-model representation, our model adopts *data/parameter samples*, instead of determinstic parameters, as meta representation, and thus can be considered as a *sample-to-sample* adaption. Meta NNSGHMC uses samples on different tasks, there is no concept of *meta samples*. Finally, PMAML fully relies on variational inference whose representation power could be restricted. In terms of model architecture, our model adopts the state-of-the-art autoregressive architectures, which can generate high-quality meta samples. Furthermore, our model adopts a simpler way to define the objective function, which allows gradients to directly flow back for meta-sampler optimization. It is worth noting that our framework reduces to MAML when only one meta sample is considered for sample adaption. Since our methods aim for Bayesian sampling, it is more similar to meta NNSGHMC (Gong et al., 2019); whereas MAML aims for point estimation of a model. Finally, we note that the recently proposed neural process (Garnelo et al., 2018) might also be used for meta-learning "few-shot function regression" as stated in the original paper. However, to our knowledge, no specific work has been done for this purpose.

## 2.2 THE BAYESIAN META SAMPLER

This section aims to design a meta sampler for efficiently generating meta samples. One idea is to use a nonparametric model to generate meta samples such as standard stochastic gradient MCMC (SG-MCMC) (Welling & Teh, 2011; Ma et al., 2015) and the Stein variational gradient descent (SVGD) (Liu & Wang, 2016a), where no parameters are consider in the model. However, methods under this setting are typically slow, and the generated samples are usually highly correlated. Most importantly, it would be hard to design nonparametric samplers that can share information between different tasks. As a result, we propose to learn the meta sampler with a parametric model, which is also denoted as a generator.

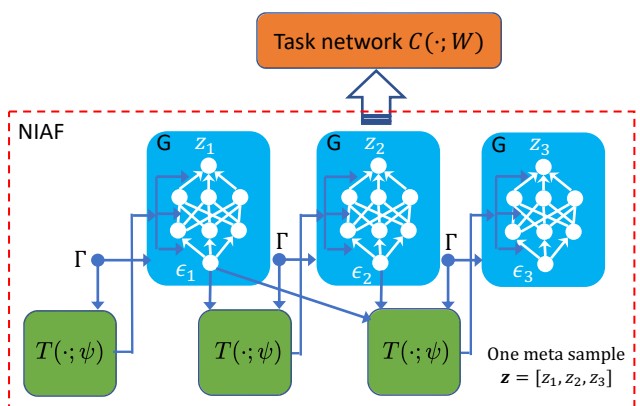

Figure 2: Roll-out architecture of the proposed NIAF as a meta sampler, consisting of a generator $G$ and an autoregressive conditional model $T(\cdot; \boldsymbol{\psi})$. The meta samples are then fed to a task network $C(\cdot; \mathbf{W})$ to encode uncertainty into specific tasks such as classification. Please see detailed descriptions in the text.

### 2.2.1 Learning parameterized meta samplers with NIAFs

There are two popular options to parameterize the meta sampler: with an explicit generator or with an implicit generator. An explicit generator parameterizes the output (*i.e.*, meta samples) as samples from an explicit distribution with a known density form such as Gaussian, thus limiting the representation power. In the following, we propose to adopt an implicit generator for the meta sampler based on neural inverse-autoregressive flows (NIAF), an inverse extension of the recently proposed NAF (Huang et al., 2018) used for density estimation. As will be seen, NIAF can incorporate task-specific information into an implicit generator and generates samples in an autoregressive manner efficiently. Finally, meta samples are used in a *task network* to encode uncertainty for specific tasks such as Bayesian classification. The architecture of the meta sampler is illustrated in Figure 2. Note the idea of using NIAF to generate network parameter is similar to the hypernetwork (Ha et al., 2016; Krueger et al., 2017). As an extention, Pradier et al. (2018) perform inference on a lower dimensional latent space. Using hypernetworks to model posterior distributions have also been studied in (Pawlowski et al., 2017; Sheikh et al., 2017).

**Neural inverse-autoregressive flows**   Directly adopting the NAF for sample generation is inappropriate as it was originally designed for density evaluation. To this end, we propose the NIAF for effective meta-sample generation. Specifically, let $z_k$ denote the $k$-th element of a sample $\mathbf{z}$ to be generated; $\tilde{\mathbf{z}}$ denotes a sample from last round; $\Gamma$ denotes the task-specific information. In our case, we set $\Gamma \triangleq (\tilde{\mathbf{z}}, \nabla_{\tilde{\mathbf{z}}} \log p(\tilde{\mathbf{z}}))$ with $p(\cdot)$ denoting the target distribution (with possible hidden parameters). NIAF generates the sample $\mathbf{z} = (z_1, \cdots, z_k, \cdots)$ via an autoregressive manner, as:

$$z_k = G(\epsilon_k, \Gamma; \boldsymbol{\phi} = T(\epsilon_{1:k-1}, \Gamma; \boldsymbol{\psi})) \,, \tag{2}$$

where $\{\epsilon_k\}$ are noise samples; $G(\cdot, \cdot; \boldsymbol{\phi})$ is an invertible function (generator) parameterized by $\boldsymbol{\phi}$ and implemented as a DNN; and $T$ is an autoregressive conditioner model parameterized by $\boldsymbol{\psi}$ to generate the parameters of the generator $G$ at each step $k$, which is itself implemented as a deep sigmoidal flow or a deep dense sigmoidal flow as in (Huang et al., 2018). According to Huang et al. (2018), using strictly positive weights and strictly monotonic activation functions for $G$ is sufficient for the entire network to be strictly monotonic, thus invertible.

**Remark 1** *The original NAFs are not designed for drawing samples, as one needs the inverse function $G^{-1}$, which is not analytically solvable when $G$ is implemented as a neural network. Although it is stated in (Huang et al., 2018) that $G^{-1}$ can be approximated numerically, one needs repeated approximations, making it computationally prohibited. Our proposed NIAF is designed specifically for sample generation by directly transforming the noise with a flow-based network $G$.*

**The task network**   In addition to the NIAF, a task network might be necessary for processing specific learning tasks. In particular, if one is only interested in generating samples from some target distribution, a task network is not necessary as the meta samples will be used to adapt to task-specific samples. However, if one wants to do classification with uncertainty, the task network should be defined as a classification network such as an MLP or CNN. In this case, denoting the weights of the task network as $\mathbf{W}$, we consider the task network as a Bayesian neural network, and propose two ways of parameterization to encode uncertainty of meta samples into the task network:

- **Sample parameterization:** A sample of the weights of the task network is directly represented by a meta sample from our meta sampler, *i.e.*, $\mathbf{W} = (z_1, z_2, \cdots, z_p)$ with $p$ the parameter dimensionality.

- **Multiplicative parameterization:** Adopting the idea of multiplicative normalizing flows (Louizos & Welling, 2017a), we define an inference network for the weights as the multiplication of $\mathbf{z}$ and a Gaussian variational distribution for $\mathbf{W}$, *i.e.*, the variational distribution is defined as the following semi-implicit distribution to approximate the true posterior distribution for $\mathbf{W}$:

$$\mathbf{z} = \text{NIAF}(\{\epsilon_k\}, \Gamma), \ \ q(\mathbf{W} \,|\, \mathbf{z}) = \prod_i \prod_j \mathcal{N}(\mathbf{W}_{ij}; z_i \mu_{ij}, \sigma_{ij}^2) \,. \tag{3}$$

  Here and in the following, we consider the task network parameterized as a one-layer MLP for notation simplicity, although our method applies to other network structures; and we have used $\text{NIAF}(\{\epsilon_k\}, \Gamma)$ to denote the output of meta samples from the NIAF.

Comparing the two parameterizations, *sample parameterization* directly generates weights of the task network from the meta sampler, thus is more flexible in uncertainty modeling. However, when the task network grows larger to deal with more complex data, this way of parameterization quickly becomes unstable or even intractable due to the high dimentionality of meta samples. *Multiplicative parameterization* overcomes this issue by associating each element of a meta sample with one node of the task network, reducing the meta-sample dimensionality from $O(N_{\text{in}} \times N_{\text{out}})$ to $O(N_{\text{in}})$ with $N_{\text{in}}$ and $N_{\text{out}}$ being the input and output sizes of the task network. As a result, we adopt the multiplicative parameterization when dealing with high dimensional problems in our experiments. Efficient inference for these two cases will be described in Section 2.4. Note a recent work on NAF inference proposes to first sample from a mean-field approximating distribution, which are then transformed by an NAF to a more expressive distribution (Webb et al., 2019). However, the approach is hard to scale to very high dimensional problems, *e.g.*, posterior distributions of network parameters.

## 2.3 THE SAMPLE ADAPTER

The output of the meta sampler contains shared (meta) information of all the tasks. Task-specific samples are expected to be adapted fast from these meta samples. This procedure is called *sample adaption*. Since there are potentially a large number of tasks, learning task-wise parametric models for sample adaption is impractical. Instead of using standard nonparametric samplers such as SG-MCMC or SVGD, we propose a general Bayesian sampling framework based on optimal-transport theory (Villani, 2008) for new task-sample adaption, where back-propagation can be directly applied.

**A general Bayesian sampling framework based on optimal transport**  Let a task-specific target distribution be $p_\tau(\mathbf{z})$, indexed by $\tau$. A standard way is to adapt the samples based on a Markov chain whose stationary distribution equals $p_\tau$, *e.g.*, via SG-MCMC. However, Markov-chain-based methods might not be efficient enough in practice due to the potentially highly-correlated samples (Chen et al., 2018b). Furthermore, it is not obvious how to apply backpropagation (BP) in most of sampling algorithms. To deal with these problems, we follow Chen et al. (2018b) and view Bayesian sampling from the Wasserstein-gradient-flow perspective (discussed in Section A.2), *i.e.*, instead of evolving samples, we explicitly evolve the underlying sample density functions. We will see that such a solution allows us to train the proposed meta sampler efficiently via standard BP.

Considering our meta-learning setting. Since we aim to adapt meta samples to new tasks, it is reasonable to define the adaptation via task-wise WGFs, *i.e.*, for each task, there is a WGF with a specific functional energy and the corresponding first variation, denoted respectively as $E_\tau$ and $F_\tau \triangleq \frac{\delta E_\tau}{\delta \rho}$ with the task index $\tau$. Here $\rho$ denotes the underlying density of the samples. Consequently, $\rho$ will evolve with a variant of the PDE by replacing $E$ with $E_\tau$ in equation 9 for each task. To solve the corresponding PDE, we prove Theorem 1 based on a discrete approximation of $\rho$ with the evolved meta samples, which is termed *optimal-transport Bayesian sampling* (OT-Bayesian sampling).

**Theorem 1 (Optimal-Transport Bayesian Sampling)** *Let $\rho_t$ at time $t$ be approximated by particles, i.e., $\rho_t(\mathbf{z}) \approx \frac{1}{M} \sum_{i=1}^M \delta_{\mathbf{z}_t^{(i)}}(\mathbf{z})$, where $\{\mathbf{z}_t^{(i)}\}$ at time $t = 0$ are a set of meta samples from the meta generator*; the delta-function $\delta_a(b) = 1$ if $a = b$ and 0 otherwise. Then sample adaption for task $\tau$ can be described with the following differential equations:*

$$\mathrm{d}\, \mathbf{z}_t^{(i)} = \nabla_{\mathbf{z}} F_\tau(\mathbf{z}_t^{(i)}) \mathrm{d}t, \quad \text{for } \forall i \,. \tag{4}$$

Based on Theorem 1, particle updates for sampling can be obtained by solving equation 4 numerically, *e.g.*, via the Euler scheme, as

$$\mathbf{z}_{k+1}^{(i)} = \mathbf{z}_k^{(i)} + h \nabla_{\mathbf{z}} F_\tau(\mathbf{z}_k^{(i)}) \,, \tag{5}$$

where $h$ is the stepsize. Here with a little abuse of notation but for conciseness, we use $\mathbf{z}_k^{(i)}$ to denote the numerical solution of $\mathbf{z}_t^{(i)}$ at time $t = hk$ with $k$ being the iteration. Theorem 1 and equation 5 indicate that the derivative of $E_\tau$ w.r.t. $\mathbf{z}_k^{(i)}$ can be considered as $\frac{\partial E_\tau}{\partial \mathbf{z}_k^{(i)}} = \nabla_{\mathbf{z}_k^{(i)}} F_\tau(\mathbf{z}_k^{(i)})$. This is a useful result to derive a learning algorithm for the meta sampler described in Section 2.4.

---

*We use the **bold** letter $\mathbf{z}_t^{(i)}$ (or $\mathbf{z}_k^{(i)}$) to denote the $i$-th meta sample evolved with equation 4 at time $t$ (or equation 5 at iteration $k$). This should be distinguished from the normal unbold letter $z_k$ defined in Section 2.2.1, which denotes the $k$-th element of $\mathbf{z}$.

**Energy functional design** Choosing an appropriate energy function $E_\tau$ is important for efficient sample adaptation. To achieve this, the following conditions should be satisfied: *i)* $E_\tau(\rho)$ should be convex w.r.t. $\rho$; *ii)* The first variation $F_\tau$ could be calculated conveniently. A general and convenient functional family is the $f$-divergence, which is defined, with our notation and a convex function $f : \mathbb{R} \to \mathbb{R}$ such that $f(1) = 0$, as: $E_\tau^f \triangleq \mathbb{D}_f(\rho\|p_\tau) = \int p_\tau(\mathbf{z})f\left(\frac{\rho(\mathbf{z})}{p_\tau(\mathbf{z})}\right) \mathrm{d}\,\mathbf{z}$. The $f$-divergence is a general family of divergence metric. With different functions $f$, it corresponds to different divergences including the popular KL divergence, inverse-KL divergences, and the Jensen-Shannon divergence. For more details, please refer to (Nowozin et al., 2016). A nice property of $f$-divergence is that its first variation endows a convenient form as stated in Proposition 2.

**Proposition 2** *Let $r(\mathbf{z}) \triangleq \frac{\rho(\mathbf{z})}{p_\tau(\mathbf{z})}$. The first variation of the $f$-divergence endows the following form:*

$$F_\tau^f \triangleq \frac{\delta E_\tau^f}{\delta \rho}(\rho) = f'(r) .$$

In our experiments, we focus on the KL-divergence, which corresponds to $f(r) = r \log r$. In this case, $\nabla_{\mathbf{z}} F_\tau^f(\mathbf{z}) = \nabla_{\mathbf{z}} \log p_\tau(\mathbf{z}) - \nabla_{\mathbf{z}} \log \rho(\mathbf{z})$. Since the density $\rho(\mathbf{z})$ required in evaluating $r$ is not readily available due to its implicit distribution, we follow Chen et al. (2018b) and use the meta samples $\{\mathbf{z}_k^{(i)}\}$ at the $k$-th step for approximation, resulting in

$$\nabla_{\mathbf{z}} F_\tau^f(\mathbf{z}) = \nabla_{\mathbf{z}} \log p_\tau(\mathbf{z}) - \sum_{i=1}^M \nabla_{\mathbf{z}_k^{(i)}} \kappa(\mathbf{z}, \mathbf{z}_k^{(i)}) / \sum_j \kappa(\mathbf{z}_k^{(i)}, \mathbf{z}_k^{(j)}) - \sum_{i=1}^M \nabla_{\mathbf{z}_k^{(i)}} \kappa(\mathbf{z}, \mathbf{z}_k^{(i)}) / \sum_{j=1}^M \kappa(\mathbf{z}, \mathbf{z}_k^{(j)}) \quad (6)$$

where $\kappa(\cdot, \cdot)$ is a kernel function. The number of adaptation steps $k$ should be set based on problems. For tasks that vary significantly, a larger $k$ should be chosen to ensure the quality of adapted samples. To further improve the accuracy, inspired by Chen et al. (2018b), we combine equation 6 with the first variation of SVGD, resulting in the following form at iteration $k$:

$$\nabla_{\mathbf{z}} F_\tau(\mathbf{z}) \triangleq \frac{1}{M} \sum_{i=1}^M \left[ \nabla_{\mathbf{z}_k^{(i)}} \log p_\tau(\mathbf{z}_k^{(i)}) \kappa(\mathbf{z}, \mathbf{z}_k^{(i)}) + \nabla_{\mathbf{z}_k^{(i)}} \kappa(\mathbf{z}, \mathbf{z}_k^{(i)}) \right] + \lambda \nabla_{\mathbf{z}} F_\tau^f(\mathbf{z}) , \quad (7)$$

where $\lambda \geq 0$ is a hyperparameter to balance the two terms.

## 2.4 MODEL TRAINING AND SAMPLE GENERATION

We first describe how to train the proposed model under the two kinds of parameterization of the task network defined in Section 2.2.1.

**Training in the sample-parameterization setting** In this case, one only needs to optimize the conditional model $T(\cdot; \boldsymbol{\psi})$ as all parameters of other networks are directly generated. Specifically, because the energy functionals for each task $\tau$ are designed so that the minima correspond to the target distributions, the objective thus can be defined over the whole task distribution $p(\tau)$ as:

$$\min \mathcal{L} \triangleq \mathbb{E}_{\tau \sim p(\tau)} \left[ E_\tau(\rho) \right] .$$

For notation simplicity, we will not distinguish among the adapted samples (*i.e.*, $\mathbf{z}_k^{(i)}$) for different tasks. Since the only parameter is $\boldsymbol{\psi}$ in the autoregressive conditioner model $T$ (see Figure 2, and note the parameters $\boldsymbol{\phi}$ and $\mathbf{W}$ for the meta generator and task network do not need to be learned as they are the outputs of $T$ and $G$, respectively), its gradient can be directly calculated using chain rule:

$$\frac{\partial \mathcal{L}}{\partial \boldsymbol{\psi}} = \mathbb{E}_{\tau \sim p(\tau)} \left[ \sum_i \left( \frac{\partial \mathbf{z}_k^{(i)}}{\partial \boldsymbol{\psi}} \right)^T \frac{\partial E_\tau}{\partial \mathbf{z}_k^{(i)}} \right] \overset{(*)}{=} \mathbb{E}_{\tau \sim p(\tau)} \left[ \sum_i \left( \frac{\partial \mathbf{z}_k^{(i)}}{\partial \boldsymbol{\psi}} \right)^T \nabla_{\mathbf{z}_k^{(i)}} F_\tau(\mathbf{z}_k^{(i)}) \right] , \quad (8)$$

where "$\overset{(*)}{=}$" follows by the results from Section 2.3 and $\nabla_{\mathbf{z}_k^{(i)}} F_\tau(\mathbf{z}_k^{(i)})$ is readily calculated with equation 7; the term $\frac{\partial \mathbf{z}_k^{(i)}}{\partial \boldsymbol{\psi}}$ can be calculated by standard BP over the meta-sampler network (NIAF).

**Training in the multiplicative-parameterization setting** In this case, two sets of parameters are to be learned, $\boldsymbol{\psi}$ and $\mathbf{W}$. For $\boldsymbol{\psi}$, one can optimize $\boldsymbol{\psi}$ by adopting the same update equation as in the sample-parameterization setting. For $\mathbf{W}$, we follow Louizos & Welling (2017a) and adopt variational

inference. Specifically, we first augment the task network with an auxiliary network with a conditional likelihood $r_\theta(\mathbf{z} \,|\, \mathbf{W}) \triangleq \prod_k \mathcal{N}(z_k | \tilde{\mu}_k, \tilde{\sigma}_k^2)$, parameterized by $\theta \triangleq \{\tilde{\mu}_k, \tilde{\sigma}_k\}$. Based on Ranganath et al. (2016); Louizos & Welling (2017a); Touati et al. (2019), with the inference network defined in (3), and writing the implicit distribution of $\mathbf{z}$ as $\tilde{q}_\psi(\mathbf{z})$ and the Gaussian prior distribution of $\mathbf{W}$ as $p(\mathbf{W})$, we arrive at the following ELBO:

$$\mathcal{L}(\theta, \phi, \psi) \triangleq \mathbb{E}_{q_\phi(\mathbf{z}, \mathbf{W})} \left[\log p(y| \mathbf{x}, \mathbf{W}, \mathbf{z}) + \log p(\mathbf{W}) + \log r_\theta(\mathbf{z} \,|\, \mathbf{W}) - \log q_\phi(\mathbf{W} \,|\, \mathbf{z}) - \log \tilde{q}_\psi(\mathbf{z})\right]$$

Different from Louizos & Welling (2017a), we only update $(\theta, \phi)$ by optimizing the above ELBO; while leave the update of $\psi$ with the gradient calculated in equation 8, which reflects gradients of samples from the NIAF. Note also that in the meta learning setting, the task network needs to be adapted for new tasks. This can be done by standard MAML with the above ELBO as the new-task objective. The meta training and testing algorithms are illustrated in the Appendix B. A similar ideas of multiplicative parametrization was proposed recently in (Kristiadi & Fischer, 2019), which used a compound density network to quantify the predictive uncertainty.

**New-task sample generation**    After training, samples for a new task can be directly generated by feeding the task information $\Gamma$ and some noise to the meta sampler depicted in Figure 1. Typically, one needs to run a few numbers of sample-adaption steps to generate good samples for new tasks. Notably, the number of sample-adaption steps required to obtain good accuracy will be shown much less than simply starting a sampler from scratch in the experiments.

## 3    EXPERIMENTS

We conduct a series of experiments to evaluate the efficiency and effectiveness of our model, and compare it with related Bayesian sampling algorithms such as SGLD, SVGD and SGHMC. The main compared algorithms for meta learning include the PMAML (Finn et al., 2018), Amortized Bayesian Meta-Learning (ABML) (Ravi & Beatson, 2019), and NNSGHMC (Gong et al., 2019), a recently proposed meta SG-MCMC algorithm. Inspired by Finn et al. (2017), we denote our algorithm *distribution agnostic meta sampling* (DAMS). Implementation details are given in Appendix E. Our code is made available at: https://github.com/zheshiyige/meta-sampling.git.

### 3.1    EVALUATIONS OF OT-BAYESIAN SAMPLING

We first demonstrate our proposed NIAF-based sampler is able to generate more effective samples compared to the popular Bayesian algorithms such as SVGD, SGLD and SGHMC, in a non-meta-sampling setting. To this end, we apply standard Bayesian Logistic Regression (BLR) on several real datasets from the UCI repository: Australian (15 features, 690 samples), German (25 features, 1000 samples), Heart (14 features, 270 samples). We perform

Table 1: Accuracy of BLR with different samplers.

|  | Australian | German | Heart |
|---|---|---|---|
| SVGD | 88.7 | 74.5 | 78.7 |
| SGLD | 88.4 | 75.2 | 81.5 |
| SGHMC | 88.7 | 77.0 | 88.6 |
| DAMS with MLP | 85.5 | 75.0 | 87.0 |
| DAMS with IAF | 87.0 | 75.5 | 88.9 |
| DAMS with NIAF | **89.1** | **77.5** | **90.7** |

posterior sampling for BLR using our proposed sampler, as well as SVGD, SGLD, SGHMC. For a more detailed investigation of different components in our model, we also test the generator with different architectures, including generators with MLP (DAMS with MLP), IAF (DAMS with IAF), and NIAF (DAMS with NIAF). We follow Liu & Wang (2016a) and apply Gaussian priors for the parameters $p_0(w|\alpha) = \mathcal{N}(w; \mathbf{0}, \alpha^{-1}\,\mathbf{I})$ with $p_0(\alpha) = \text{Gamma}(\alpha, 1, 0.01)$. A random selection of 80% data are used for training and the remaining for testing. The testing accuracies are shown in Table 1. It is observed that DAMS with NIAF achieves the best performance in terms of accuracy. The results also indicate the effectiveness and expressiveness of the proposed NIAF architecture in the OT-Bayesian sampling framework.

### 3.2    EVALUATIONS OF META-SAMPLE ADAPTABILITY

In this set of experiments, we aim to demonstrate the excellent meta-sample adaptability of our meta-sampling framework in different tasks. An additional synthetic experiment on meta posterior adaption is presented in Section D.3 of the Appendix.

**Gaussian mixture model**    We first conduct experiments to meta-sample several challenging Gaussian mixture distributions. We consider mixtures of 4, 6 and 20 Gaussians. Detailed distributional forms are given in the Appendix. To setup a meta sampling scenario, we use 2, 3 and 14 Gaussian components with different means and covariance, respectively, for meta-training of the meta sampler.

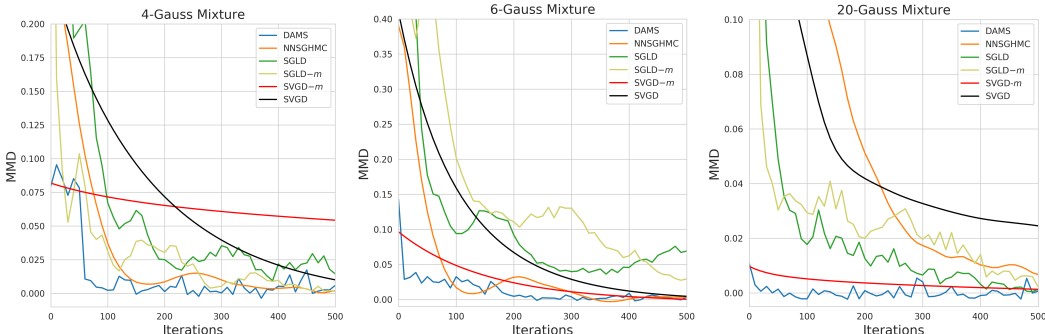

Figure 3: Convergence of meta-sampling different Gaussian mixture models. SGLD-$m$ and SVGD-$m$ are SGLD and SVGD with the same initialization as DAMS. Other samplers are initialized randomly.

After training, meta samples are adapted to samples from a target Gaussian mixture by following the new-task-sample-generation procedure described in Section 2.4. We plot the convergence of 1000 meta samples to a target distribution versus a number of particle (sample) updates (iterations), measured with the maximum mean discrepancy (MMD) evaluated by samples. For a fair comparison, we use the same number of samples (particles) to evaluation the MMD. The results are shown in Figure 3. It is clear that our proposed meta sampler DAMS converges much faster and better than other sampling algorithms, especially on the most complicated mixture of 20-Gaussians. The reason for the fast convergence (adaption) is partially due to the learned meta sampler, which provides good initialization for sample adaption. This is further verified by inspecting how samples in the adaption process evolve, which are plotted in Figure 10, 11 and 12 in the Appendix.

**Meta Sampling for Bayesian Neural Networks** Finally, we test the proposed DAMS for meta sampling of BNNs on MNIST and CIFAR-10 (Krizhevsky, 2009). We follow the experimental setting in (Gong et al., 2019), and split the MNIST and CIFAR10 dataset into two parts for meta training and testing (sample adaption), respectively. As we focus on fast adaptation, we show the accuracy within 200 iterations. To deal with the high-dimensionality issue, we adopt the method of multiplicative parameterization proposed in Section 2.2.1.

We randomly pick 5 classes for training, and the remaining classes for testing. A BNN is trained only on the training data for meta sample (weights of the BNN) generation, with each sample corresponding to a meta BNN. In testing, the meta BNNs are adapted based on the testing data. For the MNIST dataset, we parameterize a BNN as a CNN with two convolutional layers followed by a fully connected layer with 100 hidden units. The kernel and filter sizes of the two conv layers are 3 and 16, respectively. A similar architecture is applied for the CIFAR10 dataset, but with 16 and 50 filters whose kernel sizes are 7 and 5 for the two convolutional layers, respectively. The hidden units of the fully connected layer is 300.

*a) Adaptation efficiency*: For this purpose, we compare our model with NNSGHMC (Gong et al., 2019), as well as with a non-meta-learning method to train from scratch. To demonstrate the effectiveness of our NIAF structure for adaptive posterior sampling, we also compare it with the simple conditional version of MNF (Louizos & Welling, 2017a). Figure 4 plots the learning curves of testing accuracy versus the number of iterations. It is clearly seen that our DAMS adapts the fastest to new tasks, and is able to achieve the highest classification accuracy on all cases due to the effectiveness of uncertainty adaption. To further demonstrate the superiority of DAMS over NNSGHMC, we list the test accuracy at different adaptation steps in Table 2. The results clearly show faster adaption and higher accuracy of the proposed DAMS compared to NNSGHMC. It is also interesting to see that MNF with adaptation, performs better than NNSGHMC; while our method performs better than both, demonstrating the effectiveness of the proposed NIAF architecture.

*b) Sample efficiency*: To demonstrate sample efficiency of our framework, we compare it with both NNSGHMC and the standard Bayesian learning of DNNs with SGHMC. To this end, we use the first five classes of image on CIFAR10 as training data to learn the meta sampler and we randomly select 5%, 20%, 30% of the training data in the remaining image classes of the test task to adapt the meta sampler. Figure 5 shows the corresponding test accuracies for different settings. It is observed

Table 2: Accuracy (%) for adaptation efficiency.

|  | Approach | CIFAR | MNIST |
|---|---|---|---|
| 20 steps | NNSGHMC | 23.8 | 28.6 |
| 20 steps | MNF | 42.3 | 59.4 |
| 20 steps | DAMS-NIAF | **42.7** | **60.7** |
| 50 steps | NNSGHMC | 47.2 | 63.6 |
| 50 steps | MNF | 59.8 | 75.5 |
| 50 steps | DAMS-NIAF | **61.9** | **78.5** |
| 100 steps | NNSGHMC | 58.1 | 80.1 |
| 100 steps | MNF | 67.2 | 85.8 |
| 100 steps | DAMS-NIAF | **69.5** | **88.0** |
| 200 steps | NNSGHMC | 71.7 | 88.6 |
| 200 steps | MNF | 72.7 | 90.8 |
| 200 steps | DAMS-NIAF | **75.2** | **92.6** |

Table 3: Uncertainty adaptation evaluation of Bayesian meta sampling for BNNs in terms of negative log-likelihood.

|  | Approach | accuracy | NLL/100 |
|---|---|---|---|
| 20 steps | SGHMC | 32.3 | 79.22 |
| 20 steps | DAMS-NIAF | **42.7** | **68.98** |
| 50 steps | SGHMC | 39.2 | 75.34 |
| 50 steps | DAMS-NIAF | **61.9** | **57.82** |
| 100 steps | SGHMC | 46.5 | 67.95 |
| 100 steps | DAMS-NIAF | **69.5** | **46.49** |
| 200 steps | SGHMC | 59.1 | 51.79 |
| 200 steps | DAMS-NIAF | **75.2** | **36.53** |

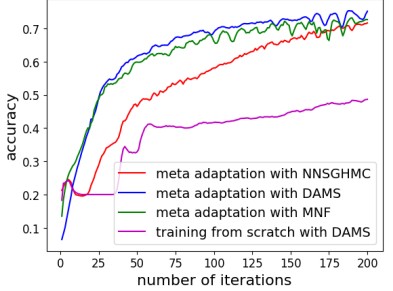 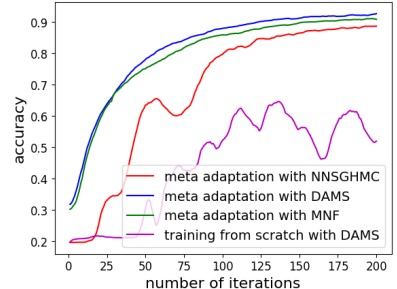

Figure 4: Adaptation efficiency in terms of testing accuracy on CIFAR10 (left) and MNIST (right).

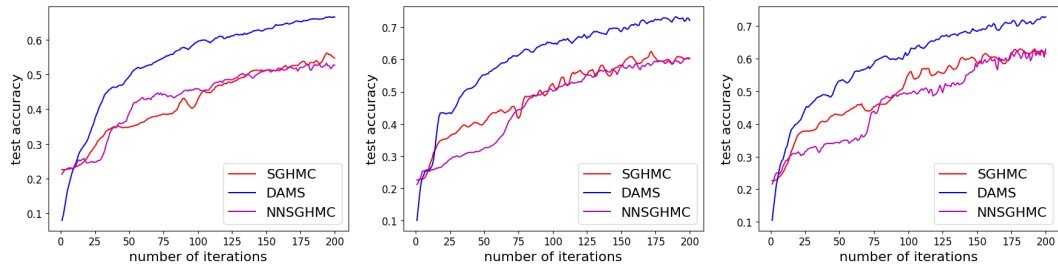

Figure 5: Sample efficiency evaluation with 5%, 20%, 30% of training data on CIFAR10. (see text).

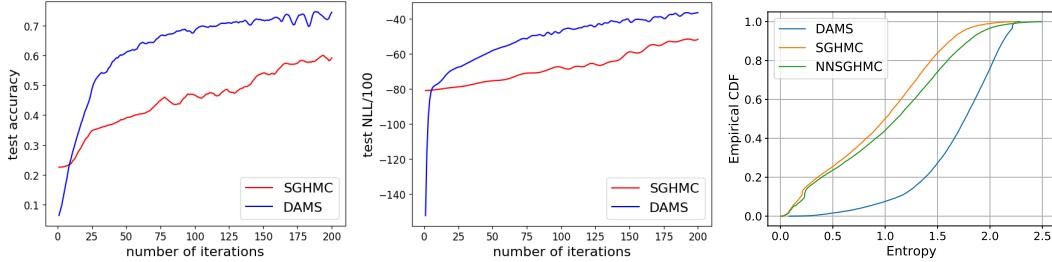

Figure 6: Testing accuracy (left, CIFAR10), uncertainty adaptation evaluation in terms of negative log-likelihood (middle, CIFAR10) and entropy of out-of-distribution prediction (right) on notMNIST.

that ours, the adaptation-based methods, obtain higher accuracies than the non-adaptive method of SGHMC. Furthermore, our method achieves the best sample efficiency among others.

*c) Uncertainty evaluation*: Finally, we ablate study the uncertainty estimation of our, the standard SGHMC and NNSGHMC models in terms of negative log-likelihood, also show the corresponding test accuracy. The results on the CIFAR10 dataset are shown in Figure 6 and Table 3. We follow Louizos & Welling (2017a) to evaluate uncertainty via entropy of out-of-sample predictive distributions (Figure 6 (right)). We observe that uncertainty estimates with DAMS are better than others, since the probability of low entropy prediction is much lower than others. Details are given in Section D.4 of the Appendix.

Table 4: Testing accuracy of Mini-ImageNet Few Shot Classification. "N/A" means results not reported.

| Mini-Imagenet 5-way Few Shot Classification | | |
|---|---|---|
| Approach | Accuracy | |
| | 1-shot | 5-shot |
| MAML | 48.70± 1.84 % | 63.11± 0.92 % |
| PMAML | 50.13 ± 1.86 % | N/A |
| ABML | 45.0 ± 0.60 % | N/A |
| MAML-SGHMC | 50.18 ± 0.50 % | 65.94 ± 0.48 % |
| MAML-SGLD | 50.14 ± 0.51 % | 65.12 ± 0.47 % |
| DAMS-SGLD | 50.86 ± 0.48 % | 66.23 ± 0.45 % |
| DAMS-NIAF | **51.43 ± 0.49** % | **66.84 ± 0.45** % |

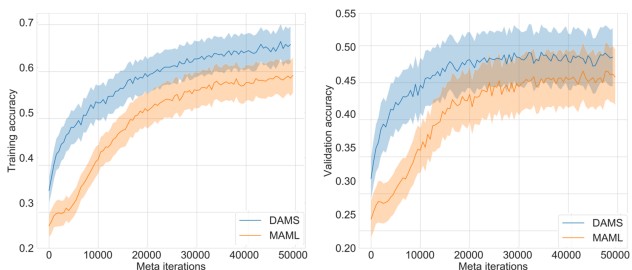

Figure 7: Learning curves of 5-way 1-shot classification.

### 3.3 META SAMPLING FOR BAYESIAN FEW-SHOT LEARNING

**DAMS for few-shot image classification** We apply our method on two popular few-shot image-classification tasks on the Mini-Imagenet dataset, consisting of 64, 16, and 20 classes for training, validation and testing, respectively. We compare our method with MAML and its variants with uncertainty modeling, including the Amortized Bayesian Meta-Learning (ABML) (Ravi & Beatson, 2019) and Probabilistic MAML (PMAML) (Finn et al., 2018). To get a better understanding of each component of our framework, we also conduct an ablation study with three variants of our model: MAML-SGLD, MAML-SGHMC and DAMS-SGLD. MAML-SGLD and MAML-SGHMC correspond to the variants where SGLD and SGHMC are used to sample the parameters of the classifier, respectively; and DAMS-SGLD replaces the WGF component of DAMS-NIAF with SGLD. Follow the setting in (Finn et al., 2017; 2018), the network architecture includes a stacked 4-layer convolutional feature extractor, followed by a meta classifier with one single fully-connected layer using the multiplicative parameterization. Testing results are presented in Table 4. With our method, we observe significant improvement of the classification accuracy at an early stage compared with MAML. The learning curves are plotted in Figure 7, further demonstrating the superiority of our method, which can provide an elegant initialization for the classification network. Finally, from the ablation study, the results suggest both the NIAF and the WGF components contribute to the performance gain obtained by our method.

**Meta sampling for reinforcement learning** We next adapt our method for meta reinforcement learning. We test and compare the models on the same MuJoCo continuous control tasks (Todorov et al., 2012) as used in (Finn et al., 2017), including the goal velocity task and goal direction task for cheetah robots. For a fair comparison, we leverage the TRPO-RL (Schulman et al., 2015) framework for meta updat-

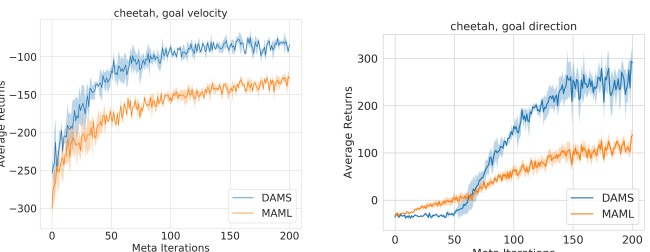

Figure 8: Locomotion comparison results for goal velocity task (left) and goal direction task (right).

ing following MAML method. Specifically, we implement the policy network with two hidden layers with ReLu activation followed by a linear layer to produce the mean value of the Gaussian policy. The first hidden layer is a fully connected layer, and we adopt the multiplicative parameterization for the second hidden layer. As shown in Figure 8, our method obtains higher rewards compared with MAML on both tasks, indicating the importance of effective uncertainty adaptation in RL.

## 4 CONCLUSION

We present a Bayesian meta-sampling framework, called DAMS, consisting of a meta sampler and a sample adapter for effective uncertainty adaption. Our model is based on the recently proposed neural autoregressive flows and related theory from optimal transport, enabling a simple yet effective training procedure. To make the proposed model scalable, an efficient uncertainty parameterization is proposed for the task network, which is trained by variational inference. DAMS is general and can be applied to different scenarios with an ability for fast uncertainty adaptation. Experiments on a series of tasks demonstrate the advantages of the proposed framework over other methods including the recently proposed meta SG-MCMC, in terms of both sample efficiency and fast uncertainty adaption.

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

## A  BACKGROUND

This section provides a review of background on Bayesian sampling, Wasserstein gradient flows and autoregressive flows.

### A.1  BAYESIAN SAMPLING

Bayesian sampling has been a long-standing tool in Bayesian modeling, with a wide range of applications such as uncertainty modeling (Li et al., 2015), data generation (Feng et al., 2017; Chen et al., 2018a) and reinforcement learning (Zhang et al., 2018). Traditional algorithms include but are not limited to Metropolis–Hastings algorithm, importance sampling and Gibbs sampling (Gelman et al., 2004). Modern machine learning and deep learning have been pushing forward the development of large-scale Bayesian sampling. Popular algorithms in this line of research include the family of stochastic gradient MCMC (SG-MCMC) (Welling & Teh, 2011; Ma et al., 2015) and the Stein variational gradient descent (SVGD) (Liu & Wang, 2016a). Recently, a particle-optimization sampling framework that unifies SG-MCMC and SVGD has also been proposed (Chen et al., 2018b), followed by some recent developments (Liu et al., 2019b;a). Generally speaking, all these methods target at sampling from some particular distributions such as the posterior distribution of the weights of a Bayesian neural network (BNN).

On the other hand, meta learning is a recently developed concept that tries to learn some abstract information from a set of different but related tasks. A natural question by considering these two is: can we design meta sampling algorithms that learns to generate *meta samples*, which can be adapted to samples of a new task-specific distribution quickly? This paper bridges this gap by proposing a mathematically sound framework for Bayesian meta sampling.

### A.2  OPTIMAL TRANSPORT AND WASSERSTEIN GRADIENT FLOWS

In optimal transport, a density function, $\rho_t$, evolves along time $t$ to a target distribution optimally, *i.e.*, along the shortest path on the space of probability measures $\mathcal{P}(\Omega)$, with $\Omega$ being a subset of $\mathbb{R}^d$. The optimality is measured in the sense that $\rho_t$ moves along the geodesic of a Riemannian manifold induced by a functional energy, $E : \mathcal{P}(\Omega) \to \mathbb{R}$, under the 2-Wasserstein distance metric. Formally, the trajectory of $\rho_t$ is described by the following partial differential equation (PDE):

$$\partial_t \rho_t = \nabla_{\mathbf{z}} \cdot \left( \rho_t \nabla_{\mathbf{z}} (\frac{\delta E}{\delta \rho_t}(\rho_t)) \right) \triangleq \nabla_{\mathbf{z}} \cdot (\rho_t \nabla_{\mathbf{z}} F) \ , \tag{9}$$

where $\nabla_{\mathbf{z}} \cdot f \triangleq \sum_i \frac{\partial}{\partial \mathbf{z}_i} f$ is the divergence operator; and $F \triangleq \frac{\delta E}{\delta \rho_t}(\rho_t)$ is called the *first variation* of $E$ at $\rho_t$ (functional derivative on a manifold in $\mathcal{P}(\Omega)$). To ensure $\rho_t$ to converge to a target distribution $p$ such as the posterior distribution of the model parameters, one must design an appropriate $E$ such that $p = \arg_\rho \min E(\rho)$. A common choice is the popular KL-divergence, $\mathsf{KL}(\rho, p)$. We will consider a more general setting in our framework presented later. Note the WGF framework equation 9 allows to view Bayesian sampling from a density-optimization perspective. For example, recent works (Chen et al., 2018b; Liu et al., 2019b;a) consider approximating $\rho_t$ with samples and evolve the samples according to equation 9. Parts of our model will follow this sampling setting.

### A.3  AUTOREGRESSIVE FLOWS AND VARIANTS

Our model relies on the concept of autoregressive flows for meta-sampler design. We review some key concepts here. More detailed comparisons are provided in the Appendix.

A normalizing flow defines an invertible transformation from one random variable $\epsilon$ to another $\mathbf{z}$. A flexible way to implement this is to define it via implicit distributions, meaning sample generation is implemented as: $\epsilon_i \sim q_0(\epsilon_i)$, $\mathbf{z}_i = G(\epsilon_i; \boldsymbol{\phi})$, where $i$ indexes elements of $\epsilon$ and $\mathbf{z}$; $G$ represents a deep neural network (generator) parameterized by $\boldsymbol{\phi}$.

The autoregressive flow (AF) parameterizes a Gaussian conditional distribution for each $z_i$, *e.g.*, $p(z_i | \mathbf{z}_{1:i-1}) = \mathcal{N}(z_i | \mu_i, \exp(\alpha_i))$, where $\mu_i = g_{\mu_i}(\mathbf{z}_{1:i-1})$ and $\alpha_i = g_{\alpha_i}(\mathbf{z}_{1:i-1})$ are outputs of two neural networks $g_{\mu_i}$ and $g_{\alpha_i}$. The sample generation process is: $z_i = \epsilon_i \exp(\alpha_i) + \mu_i$, with $\mu_i = g_{\mu_i}(\mathbf{z}_{1:i-1})$, $\alpha_i = g_{\alpha_i}(\mathbf{z}_{1:i-1})$ and $\epsilon_i \sim \mathcal{N}(0, 1)$. Instances of autoregressive flows include the

Autoregressive Flow (AF) (Chen et al., 2017) and Masked Autoregressive Flow (MAF) (Papamakarios et al., 2017). The inverse autoregressive flow (IAF) (Kingma et al., 2016) is an instance of normalizing flow that uses MADE (Germain et al., 2015), whose samples are generated as: $z_i = \epsilon_i \exp(\alpha_i) + \mu_i$, with $\mu_i = g_{\mu_i}(\epsilon_{1:i-1})$, $\alpha_i = g_{\alpha_i}(\epsilon_{1:i-1})$ and $\epsilon_i \sim \mathcal{N}(0, 1)$.

The neural autoregressive flow (NAF) (Huang et al., 2018) replaces the affine transformation used in the above flows with a deep neural network (DNN), *i.e.*, $\epsilon_t = f(\mathbf{z}_t, \boldsymbol{\phi} = T(\mathbf{z}_{1:t-1}))$, where $f$ is a DNN transforming a complex sample distribution, $p(\mathbf{z})$, to a simple latent representation $q_0(\epsilon)$. In NAF, $q_0$ is considered as a simple prior, and $f$ is an invertible function represented by a DNN, whose weights are generated by $T$, an autoregressive conditional model. Let the induced distribution of $\epsilon$ by $f$ be $p_f(\epsilon)$. $f$ is learned by minimizing the KL-divergence between $p_f(\epsilon)$ and $q_0(\epsilon)$.

### A.4 COMPARISONS BETWEEN DIFFERENT FLOWS

Note the $\mu_i$ and $\alpha_i$ are computed differently for AF and IAF, *i.e.*, previous variables $\mathbf{z}_{1:i-1}$ are used for AF and previous random noise $\epsilon_{1:i-1}$ are used for IAF. AF can be used for calculating the density $p(\mathbf{z})$ of any sample $\mathbf{z}$ in one pass of the network. However, drawing samples requires performing $D$ sequential passes ($D$ is the dimensionality of $\mathbf{z}$). Thus if $D$ is large, drawing samples will be computationally prohibited. IAF, by contrast, can draw samples and estimate densities of the generated samples with only one pass of the network. However, calculating the sample density $p(\mathbf{z})$ requires $D$ passes to find the corresponding noise $\epsilon$. The advantage of NAF is that the mapping function $f$ is much more expressive, and density evaluation is efficient. However, drawing samples is much more computationally expensive. To adopt the NAF framework for sampling, we propose the neural inverse-autoregressive flow (NIAF) in Section 2.2.1.

## B DAMS WITH ELBO AS THE OBJECTIVE

Our training algorithm in the multiplicative-parameterization setting includes updating the flow parameter and learning the task network with variational inference, which is described in Algorithm 1 and testing setting in Algorithm 2.

---

**Algorithm 1** Meta training, MAML with ELBO.

---

**Require:** $p(\mathcal{T})$: distribution over tasks;
**Require:** $\alpha, \beta$ step size hyperparameter;
  randomly initialize $\boldsymbol{\theta}, \boldsymbol{\phi}, \boldsymbol{\psi}$.
  **while** not done **do**
    Sample batch of tasks $\mathcal{T}_i \sim p(\mathcal{T})$
    **for all** $\mathcal{T}_i$ **do**
      Sample K datapoints $\mathcal{D}_\tau$ for each task $\tau$ in $\mathcal{T}_i$
      Evaluate $\nabla_{(\boldsymbol{\theta}, \boldsymbol{\phi})}\mathcal{L}(\boldsymbol{\theta}, \boldsymbol{\phi}, \boldsymbol{\psi})$ using ELBO and $\nabla_{\boldsymbol{\psi}}\mathcal{L}(\boldsymbol{\theta}, \boldsymbol{\phi}, \boldsymbol{\psi})$ with Equation 8
      Compute adapted parameters with gradient descent:
      $(\boldsymbol{\theta}'_i, \boldsymbol{\phi}'_i) = (\boldsymbol{\theta}, \boldsymbol{\phi}) - \alpha\nabla_{(\boldsymbol{\theta}, \boldsymbol{\phi})}\mathcal{L}_{\mathcal{T}_i}(\boldsymbol{\theta}, \boldsymbol{\phi}, \boldsymbol{\psi})$
      $\boldsymbol{\psi}'_i = \boldsymbol{\psi} - \alpha\nabla_{\boldsymbol{\psi}}\mathcal{L}_{\mathcal{T}_i}(\boldsymbol{\theta}, \boldsymbol{\phi}, \boldsymbol{\psi})$
      Sample datapoints $\mathcal{D}'_\tau$ for each task $\tau$ in $\mathcal{T}_i$ for the meta-update
    **end for**
    Update $(\boldsymbol{\theta}, \boldsymbol{\phi}, \boldsymbol{\psi}) = (\boldsymbol{\theta}, \boldsymbol{\phi}, \boldsymbol{\psi}) - \beta\nabla_{\boldsymbol{\theta}, \boldsymbol{\phi}, \boldsymbol{\psi}} \sum_{\mathcal{T}_i \sim p(\mathcal{T})} \mathcal{L}(\boldsymbol{\theta}'_i, \boldsymbol{\phi}'_i, \boldsymbol{\psi}'_i)$
  **end while**

---

**Algorithm 2** Meta testing, MAML with ELBO.

---

**Require:** Training data $\mathcal{D}^{tr}_\tau$ for task $\tau$;
**Require:** Meta learned initialization of $\boldsymbol{\theta}, \boldsymbol{\phi}, \boldsymbol{\psi}$.
  Evaluate $\nabla_{(\boldsymbol{\theta}, \boldsymbol{\phi})}\mathcal{L}(\boldsymbol{\theta}, \boldsymbol{\phi}, \boldsymbol{\psi})$ using ELBO and $\nabla_{\boldsymbol{\psi}}\mathcal{L}(\boldsymbol{\theta}, \boldsymbol{\phi}, \boldsymbol{\psi})$ with Equation 8
  Compute adapted parameters with gradient descent:
  $(\boldsymbol{\theta}'_i, \boldsymbol{\phi}'_i) = (\boldsymbol{\theta}, \boldsymbol{\phi}) - \alpha\nabla_{(\boldsymbol{\theta}, \boldsymbol{\phi})}\mathcal{L}(\boldsymbol{\theta}, \boldsymbol{\phi}, \boldsymbol{\psi})$
  $\boldsymbol{\psi}'_i = \boldsymbol{\psi} - \alpha\nabla_{\boldsymbol{\psi}}\mathcal{L}(\boldsymbol{\theta}, \boldsymbol{\phi}, \boldsymbol{\psi})$

---

## C   PROOF OF THEOREM 1 AND PROPOSITION 2

**Proof**  For each task $\tau$, we first write out the corresponding WGF as

$$\partial_t \rho_t = \nabla_{\mathbf{z}} \cdot (\rho_t \nabla_{\mathbf{z}} F) \ . \tag{10}$$

Note the WGF is defined in the sense of distributions Ambrosio et al. (2005), meaning that for any smooth real functions $u(\mathbf{z})$ with compact support, equation 10 indicates

$$\int_0^T \int_{\mathbb{R}^d} \partial_t u(\mathbf{z}, t) \rho_t \mathrm{d}t \mathrm{d}\,\mathbf{z} = \int_0^T \int_{\mathbb{R}^d} \nabla_{\mathbf{z}} u(\mathbf{z}) \cdot (\nabla_{\mathbf{z}} F \rho_t) \mathrm{d}t \mathrm{d}\,\mathbf{z} \ ,$$

or equivilently,

$$\frac{\mathrm{d}}{\mathrm{d}t} \int \rho_t u(\mathbf{z}) \mathrm{d}\,\mathbf{z} = \int \nabla_{\mathbf{z}} u(\mathbf{z}) \cdot \nabla_{\mathbf{z}} F \rho_t \mathrm{d}\,\mathbf{z}$$

$$\Rightarrow \frac{\mathrm{d}}{\mathrm{d}t} \mathbb{E}_{\rho_t} [u(\mathbf{z})] = \mathbb{E}_{\rho_t} [\nabla_{\mathbf{z}} u(\mathbf{z}) \cdot \nabla_{\mathbf{z}} F] \ . \tag{11}$$

Taking $\rho(t) = \frac{1}{M} \sum_{i=1}^M \delta_{(\mathbf{z}_t^{(i)})}(\mathbf{z})$, and for each particle letting $u(\mathbf{z}) = \mathbf{z}$, equation 11 then reduces to the following differential equation for each particle:

$$\frac{\mathrm{d}\,\mathbf{z}_t^{(i)}}{\mathrm{d}t} = \nabla_{\mathbf{z}_t^{(i)}} F(\mathbf{z}_t^{(i)}) \ ,$$

which is the particle evolution equation we need to solve. ∎

**Proof** [Proof of Proposition 2] First, we introduce the following result from Ambrosio et al. (2005) [page 120]:

**Lemma 3**  *Let $\mathcal{F}(\rho) \triangleq \int \tilde{f}(\rho(x)) \mathrm{d}x$, then the first variation of $\mathcal{F}$ w.r.t. $\rho$ is*

$$\frac{\delta \mathcal{F}}{\delta \rho}(\rho) = \tilde{f}'(\rho) \ .$$

In the f-divergence case, this corresponds to $\tilde{f}(\rho) = p_\tau f(\frac{\rho}{p_\tau})$. Applying Lemma 3 and using the chain rule, we have

$$\frac{\delta E_\tau^f}{\delta \rho}(\rho) = \left( p_\tau f(\frac{\rho}{p_\tau}) \right)'$$

$$= p_\tau \frac{\mathrm{d}}{\mathrm{d}\rho} \left( \frac{\rho}{p_\tau} \right) f'(\frac{\rho}{p_\tau}) = f'(\frac{\rho}{p_\tau}) = f'(r) \ ,$$

which completes the proof. ∎

## D   EXTRA EXPERIMENTAL RESULTS

This section provides extra experimental results to demonstrate the effectiveness of our proposed method.

### D.1   SYNTHETIC DISTRIBUTIONS

Analytic forms of synthetic distributions are provided below.

*Mog4*:

$$p(\mathbf{z}) = \sum_{i=1}^4 \mathcal{N}(\mathbf{z} \,|\, \boldsymbol{\mu}_i, \boldsymbol{\sigma}^2)/4$$

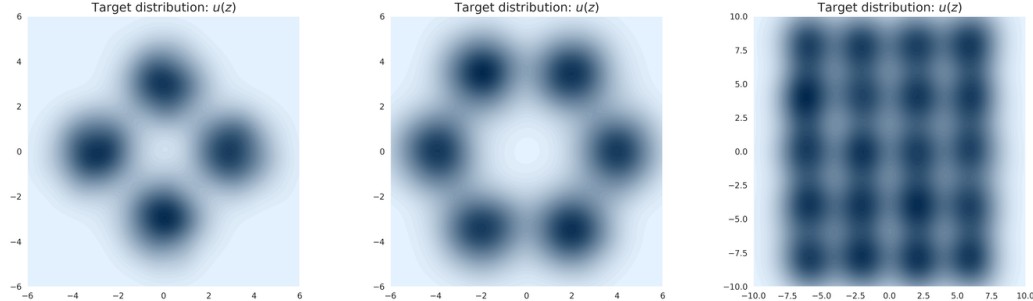

Figure 9: Densities of three Gaussian mixtures.

where $\mathbf{z} \in \mathbb{R}^2, \boldsymbol{\mu}_1 = -\boldsymbol{\mu}_2 = [3, 0], \boldsymbol{\mu}_3 = -\boldsymbol{\mu}_4 = [0, 3], \boldsymbol{\sigma} = [1, 1]$.

*Mog6*:

$$p(\mathbf{z}) = \sum_{i=1}^{6} \mathcal{N}(\mathbf{z} \,|\, \boldsymbol{\mu}_i, \boldsymbol{\sigma}^2)/6$$

where $\mathbf{z} \in \mathbb{R}^2, \boldsymbol{\mu}_i = [4\cos(i\pi/3), 4\sin(i\pi/3)], \boldsymbol{\sigma} = [1, 1]$.

*Mog20*:

$$p(\mathbf{z}) = \sum_{i=1}^{20} \mathcal{N}(\mathbf{z} \,|\, \boldsymbol{\mu}_i, \boldsymbol{\sigma}^2)/20$$

where $\mathbf{z} \in \mathbb{R}^2, \boldsymbol{\mu} = meshgrid(x, y), x = linspace(-6, 6, 2), y = linspace(-8, 8, 4), \boldsymbol{\sigma} = [1, 1]$.

The distributions are plotted in Figure 9

## D.2 EXTRA EXPERIMENTAL RESULTS ON GAUSSIAN MIXTURE MODELS

For the Gaussian mixture models, we provide extra experimental results showing how samples are evolved in the adaptation process, as illustrated in Figure 10, 11 and 12.

## D.3 META POSTERIOR ADAPTION

We apply our DAMS for fast adaptive sampling on regression tasks. We follow Huang et al. (2018) and apply DAMS to sample the posterior distribution of the frequency parameter of a sine-wave function, given only three data points. The sine-wave function is defined as $y(t) = \sin(2\pi ft)$, with a uniform prior $U([0, 2])$ and a Gaussian likelihood $\mathcal{N}(y_i; y_f(t_i), 0.125)$.

We design a meta-sampling setting to adapt the posterior distributions, $p(f|\mathcal{D})$ on the training data, to that on new test data. Specifically, meta training data

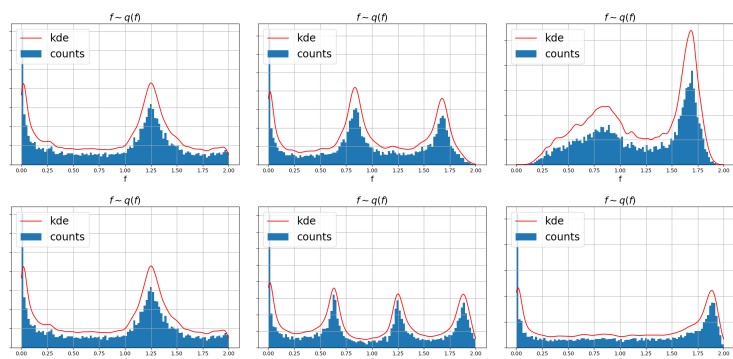

Figure 13: Meta sampling Bayesian regression for two test tasks (top and bottom). Each row plots posteriors after meta training (left), meta testing (middle, 0.8K (top) and 0.9K (bottom) iterations) and re-training (right, 0.8K (top) and 0.9K (bottom) iterations).

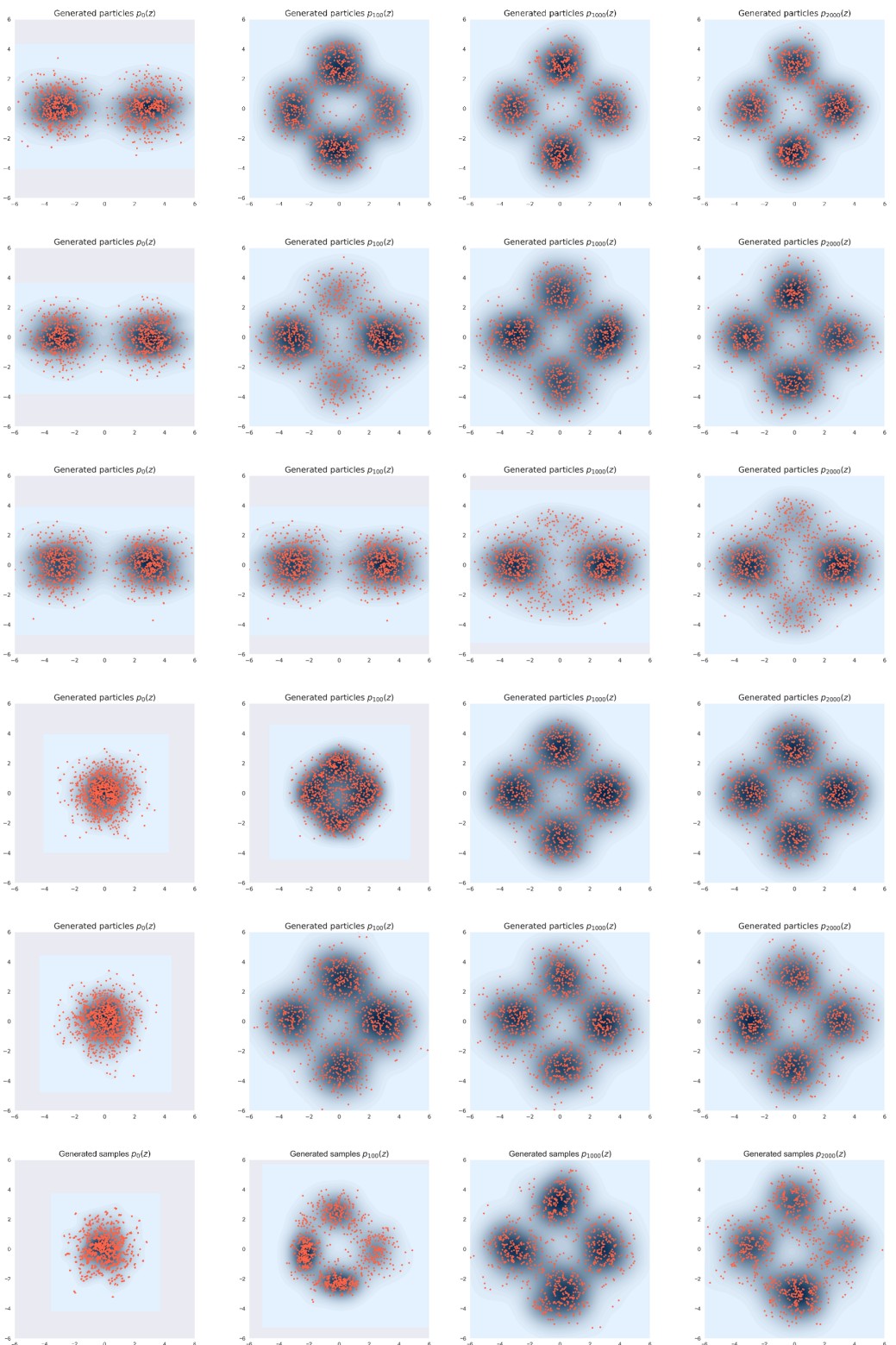

Figure 10: Comparison among different samplers on adapting to Mixture of 4-Gaussian. Top to Bottom row: DAMS, SGLD-$m$, SVGD-$m$, SVGD, SGLD and NNSGHMC

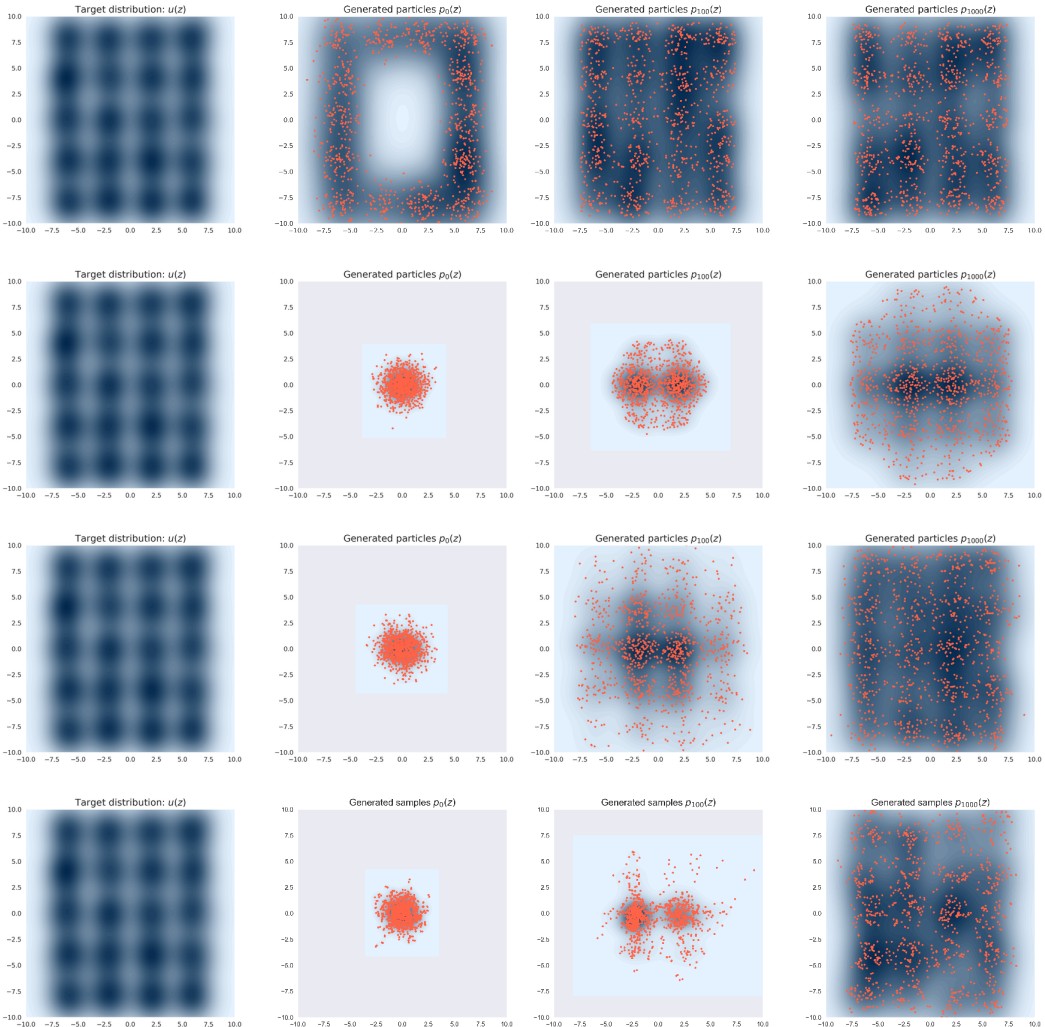

Figure 11: Comparison among different samplers on adapting to Mixture of 20-Gaussian. Top to Bottom row: DAMS, SVGD, SGLD and NNSGHMC

$(t, y)$ are $\{(0,0), (2/5,0), (4/5,0)\}$. For the first setting, meta testing consists of data $\{(0,0), (3/5,0), (6/5,0)\}$. For the second setting, meta testing consists of data $\{(0,0), (4/5,0), (8/5,0)\}$. Meta training data corresponds to a posterior with two modes of $f \in \{0.0, 5/4\}$. For the first setting, the test data corresponds to a posterior with three modes $f \in \{0.0, 5/6, 5/3\}$. For the second setting, the test data corresponds to four modes $f \in \{0.0, 5/8, 5/4, 15/8\}$. We compare our DAMS with the result of re-training from scratch with the test data. Empirical distribution with samples and kernel density estimation are plotted in Figure 13. The first setting takes about 3.4K iterations to find the three modes in the posterior with re-training, while it takes about 0.8K iterations with meta adaptation. For the second setting, it takes more than 3.6K iterations to find the four modes with training from scratch, while it is about 0.9K iterations with meta adaptation. For both test tasks, the sampler with re-training miss at least one mode compared with the meta sampler adaptation with the same number of iterations. We can see that DAMS can adapt the training posterior to the test posterior much faster than re-training from scratch due to effective uncertainty adaption, obtaining more than 3X speedups.

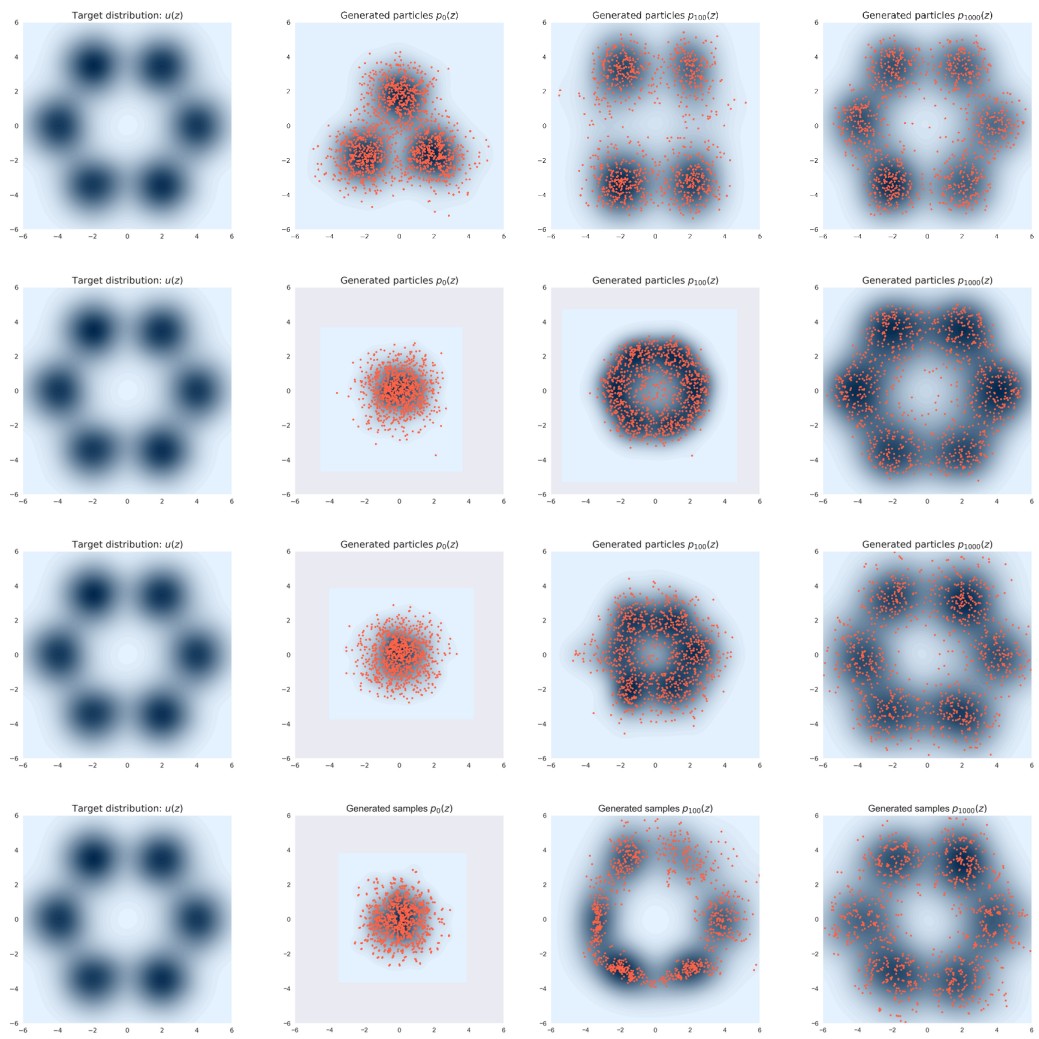

Figure 12: Comparison among different samplers on adapting to Mixture of 6-Gaussian. Top to Bottom row: DAMS, SVGD, SGLD and NNSGHMC

## D.4 UNCERTAINTY EVALUATION VIA ENTROPY OF OUT-OF-SAMPLE PREDICTIVE DISTRIBUTIONS

We show the predictive uncertainty of DAMS compared to SGHMC and NNSGHMC by exploring the posterior of neural parameters, we estimate the uncertainty for out-of-distribution data samples Lakshminarayanan et al. (2017). We train different algorithms on the MNIST dataset, and estimate the entropy of the predictive distribution on the notMNIST dataset Bulatov (2011). We follow Louizos & Welling (2017a) and use the empirical CDF of entropy to evaluate the uncertainty. Since the probability of observing a high confidence prediction is low, curves that are nearer to the bottom right of the figure estimates uncertainty better. The predictive distribution of the trained model is expected to be uniform over the notMNIST digits as the samples from the dataset are from unseen classes. The BNN is a CNN with 16 and 50 filters whose kernel sizes are 5 and 5 for the two convolutional layers, respectively. The hidden units of the fully connected layer is 300. The uncertainty evaluation results are shown in the right plot of Figure 6.

# E   IMPLEMENTATION DETAILS

For all the experiments, we follow Liu & Wang (2016b) and use RBF kernel, whose the bandwidth is calculated as $h = med^2/logn$ with $med$ the median of the pairwise distance between all the samples.

## E.1   SYNTHETIC DISTRIBUTIONS

The NIAF is fixed with two layers of deep sigmoidal flows (Huang et al., 2018). A small neural network of 1 hidden layers with 16 sigmoid units is used. The model is trained with Adam (Kingma & Ba, 2015) with a learning rate of 0.005. We use 1000 particles to approximate the distribution. We set the weight $\lambda$ in the WGF to be 1e-4. The meta sampler is trained for 1000 iterations. After that, the meta sampler is adapted to the target distribution for 500 iterations. The models are trained in the sample-parameterization setting.

## E.2   BAYESIAN LOGISTIC REGRESSION

For all the three UCI datasets, the NIAF is fixed with two layers of deep sigmoidal flows Huang et al. (2018). The model is trained with Adam Kingma & Ba (2015). We set the batch size to be 100, learning rate to be 0.005. We use 100 particles to approximate the distribution. We set the weight $\lambda$ in the WGF to be 1e-4. The hidden layer of each MADE has 64 hidden unit. We use use small neural network of 1 hidden layers with 16 sigmoid units. The accuracy is selected as the best results. The models are trained in the sample-parameterization setting.

## E.3   CIFAR AND MNIST

The NIAF is fixed with two layers of deep dense sigmoidal flow (Huang et al., 2018). The model is trained with Adam (Kingma & Ba, 2015). We set the batch size to be 128, and the learning rate to be 0.002. We set the weight $\lambda$ in the WGF to be 1e-4. The hidden layer of each MADE has 64 hidden unit. We use use a small neural network of 1 hidden layers with 16 sigmoid units. For the auxiliary network, we follow Louizos & Welling (2017b) and use the stable updates of masked RealNVP Dinh et al. (2017) in Inverse Autoregressive Flow (IAF) Kingma et al. (2016) with 50 hidden units of length two to parametrize $r_{\boldsymbol{\theta}}(\mathbf{z} \mid \mathbf{W})$. The meta sampler is trained for 10 epochs. After that, the meta sampler is adapted to unseen image classes for 200 iterations. For NNSGHMC and our DAMS, 20 meta samples are used for testing evaluation.

## E.4   FEW SHOT LEARNING

For few shot learning, we adopt the open-source Pytorch MAML implementation `https://github.com/AntreasAntoniou/HowToTrainYourMAMLPytorch`. We incorporate the proposed DAMS-NIAF as a meta classifier. The whole training procedure consists of two steps: 1) Train the model to reach the highest validation accuracy. As shown in Figure 7, a fast adaptive classifier that interacts well with the overall model is acquired. 2) Keep all the flow parameters fixed and continue training until 100 epochs.

## E.5   META REINFORCEMENT LEARNING

We evaluate our method on goal velocity task and goal direction task for cheetah robots. We leverage HalfCheetahDir-v1 environment. We set the fast learning rate (learning rate for the inner loop 1-step gradient update of MAML) to be 0.1, max-kl (maximum value for the KL constraint in TRPO) to be 0.001, task batch size (batch size for each individual task) to be 20, meta batch size (number of tasks per batch) to be 40. For policy network, we have two layers, the first layer is fully connected layer and the second layer is multiplicative parametrization layer following a relu activation function.

