# OpenReview forum: "Bayesian Meta Sampling for Fast Uncertainty Adaptation"
_ICLR.cc/2020/Conference — Accept (Poster)_

### Official Review · AnonReviewer3 · 2019-10-21
**Official Blind Review #3**

**Rating:** 6

**Review:**

Thank you for an interesting read.

As far as I understand, this paper presents DAMS which is a MAML-like algorithm but applied to posterior sampling. The idea is the following:
1. Construct a meta-sampler that generates good proposals/initial samples for task-specific samplers;
2. Train the meta-sampler so that the task-specific posterior sampling converges faster to the target distribution.

The meta-sampler is designed as an inverse version of the neural auto-regressive flow (NIAF), and the task-specific sampler is based on the Wasserstein gradient flow (WGF).

======= novelty ======
The method is novel:
1. Probabilistic/Bayesian understanding of few-shot learning/meta learning has been proposed, but to date variational inference is the main inference engine used in the literature. This paper provides a nice complement by considering fast adaptation of posterior sampling.
2. The meta-level parameterisation method is indeed different from (probabilistic) MAML, in the sense that the initial parameters/samples z is generated from a neural network conditioned on a task, instead of using a shared initialisation across tasks. This is more inline with approaches such as hyper-networks, and I haven't seen many MAML-like approaches doing that. The meta-sampler architecture is new but improved upon NAF so I consider the architectural novelty to be minor.


======= significance ======
The experimental section contains many results, with the two main category as (a) comparisons between DAMS and existing sampling methods on Bayesian inference tasks; and (b) comparisons to MAML on few-shot learning tasks. Compared with the baselines, DAMS achieves significantly better results, which is a good sign.

However, DAMS as a whole pipeline has a lot of components, and it is not clear to me which part is the main driving force. So I think the following ablation studies will be very helpful:
1. To see whether it is necessary to use NIAF, one can replace NIAF with a set of learned particles shared across tasks (i.e. make \psi = {z^(1), ..., z^(N)} which is in similar spirit as MAML).
2. To see whether WGF brings in significant improvement in DAMS, one can replace WGF in sample adapter with SVGD or SG-MCMC method such as SGHMC/preconditioned SGLD. A comparison between e.g. SGHMC vs DAMS-SGHMC vs DAMS-WGF will be helpful for this ablation.
3. It seems to me the meta-sampler requires running a few step of NAIF updates (\Gamma info contains previous sample and the gradients). How many steps are required here? A detailed analysis will be useful. If a lot of steps are required, the "fast adaptation" in meta-testing is not really the case, as both meta-sampling and sample adaptation requires evaluating gradients.
4. The multiplicative normalising flow for BNN approximate posterior adds in another layer of complexity, as the method also perform meta-learning on this variational distribution. So a baseline which removes the NIAF part (i.e. also learn particles for masks z, see point 1) on few-shot learning tasks would be useful. As far as I understand this baseline approach is different from ABML/PMAML that has been reported in the paper.

======= clarity =======
The presentation needs to be improved.
To me, section 3 seems to be overwhelmed by details, e.g. the long discussion of task networks and WGF. For example, to what extent do the WGF construction details matter for understanding the whole DAMS pipeline? I think the general idea works for any valid posterior sampler in the fast adaptation step.
I would suggest the following structure for section 3 instead:
1. write down the whole pipeline in a more abstract way, e.g. say the meta-sampler is any generator conditioned on task information, and the sample adaptation as generic posterior sampling;
2. discuss the training algorithm, what are the loss functions, etc;
3. discuss in meta-testing how DAMS is deployed;
3. discuss the detail implementation of the meta-sampler and sample adaptation method.

**Experience Assessment:**

I have published one or two papers in this area.

**Review Assessment: Checking Correctness Of Derivations And Theory:**

I carefully checked the derivations and theory.

**Review Assessment: Checking Correctness Of Experiments:**

I assessed the sensibility of the experiments.

**Review Assessment: Thoroughness In Paper Reading:**

I read the paper at least twice and used my best judgement in assessing the paper.

---

> ### Author Response · Authors · 2019-11-15
> **Response to Reviewer3**
>
> We appreciate your support for our paper. We summarize your major concerns below and will address them in the rebuttal and our revision.
>
> 1. Ablation studies.
>
> A: To address your concerns, we have conducted a number of extra experiments, summarized as:
> We replace NIAF with a set of learned particles shared across tasks. Specifically, we compare to the versions of MAML with SGLD, SGHMC. We make comparison between MAML-SGLD vs DAMS-SGLD vs DAMS-WGF
>
> The results are shown in Table 4. From the results, we can conclude that (1) it is beneficial to use NIAF; (2) WGF brings in significant improvement in DAMS.
>
> 2. It seems the meta-sampler requires running a few step of NAIF updates. How many steps are required here?
>
> A: Our meta-sampler is parameterized by a DNN (specifically an NIAF). In testing, we only need to forward through the NIAF “once” to generate meta samples, thus is very efficient. In fact, once would need several adaptation steps in the “sample adapter” though, which we usually set it to be 1 to 5 steps.
>
> 3. Paper restructure.
>
> A: Thanks for the suggestion. We will surely consider your suggestions to restructure the paper. Considering the time constraint and many extra experimental results added in, we will make corresponding changes in the final version of the paper.

---

### Official Review · AnonReviewer1 · 2019-10-24
**Official Blind Review #1**

**Rating:** 6

**Review:**

Summary of paper:
The authors propose a neural sampler for probabilistic models in the meta-learning setting.
Their main claim is that their model captures uncertainty in samples better than competing methods and does so at lower cost.
In particular, they propose a scheme which separates sampling variables for a task tau into two components: a meta-sampler and a sample adapter.
The meta-sampler intuitively plays the role of  a learned conditional distribution over the target variables.
The sample adapter is a sampler which is seeded with samples from the meta-sampler and moves them towards the desired data-distribution based on a technique called optimal-transport Bayesian sampling.
Crucially, the meta-sampler is based on neural inverse autoregressive flows to have adequate representational capacity.
The authors use the proposed algorithm (denoted DAMS for distribution agnostic meta sampling) for a variety of tasks:
First, parameters for logistic regression models are generated and evaluated on various UCI tasks.
Second, meta-learning over Gaussian mixture models (GMMs) is performed.
Third, posterior adaption in a toy regression task where the frequency parameter of a sine wave is estimated.
Last, the authors train neural networks with DAMS and test both classification accuracy as a function of test-time inference for test datasets of held out classes on cifar10, mnist and a few-shot training example on Mini-Imagenet, while also testing their method on meta-reinforcement learning Mujoco tasks.


Main Comments to authors:

Pros:
-interesting combination of techniques (IAF flows and WGF/Stein inference) to do meta-sampling
-empirically appealing results as pertaining to raw performance metrics like accuracy

Weaknesses:
- The evaluation is focused on #of steps during testing, but not on # of datapoints required for eval on new domains.
-> this is only half of what we care about when saying a sampler is "fast". The other half would be sample efficiency, which is typically the main motivation for Bayesian models and accurate uncertainty estimates, for instance when performing Bayesian Optimization. In fact, when making claims about uncertainty estimates as the goal of the paper, it would be most interesting to see how much test data the method needs to ingest before producing calibrated estimates. The method as currently presented only evaluates speed in terms of computation, but ignores sample efficiency entirely. As such, it is unclear from the given experiments to evaluate the main claim of the paper: that DAMS improves uncertainty adaptation.
-> the uncertainty is barely evaluated except in the low-d and toy sine wave illustration, which probably can be done equally well or better with regular posterior inference on D = D_train union D_test, i.e. using HMC. My suggestion to the authors would be to consider comparing to regular Bayesian inference (i.e. Monte Carlo/HMC/SVI) based on train_data and test data to compare to a ground truth estimate they might want.

-In Sec. 3.3 Eq. 6 and 7 a kernel is used and then not discussed much further. Kernels on high dimensional data (such as the weights of a NN)  are problematic to be used due to the curse of dimensionality. For the meta-sampler, even in the case of the multiplicative parametrization which lowers dimensionality, this would indicate that the kernel part of the objective might not be doing much work at all as in high enough dimensions all distances become even. If that were to be the case, the model might just look for the mode in any high-d example instead of actually sampling from a posterior and producing uncertainty estimates. Any empirical analysis and discussion on this is entirely missing here, unfortunately. As presented, the reader just has to accept that the objective functions make sense because the final product of putting all of the components together produces high accuracies. I would appreciate more details and careful analysis.

-Please also show the performance of meta-sampler with and without sample adapter in this case to clarify the effects of performing sample adaptation versus just using the meta-sampler, as this also is never compared in the paper. I.e. how good of a conditional model is the autoregressive flow? How much work does the sampler have to do? Would another conditional model do as well or worse? Why this choice of conditional model in particular if in the end sampling is put on top of it?

-The ELBO in Sec.3.4 involves an inference network over NN parameters (or potentially latent Zs per feature when using multiplicative parametrization). This object is highly nontrivial and not analyzed in terms of performance at all here.  Inference networks over neural networks are hard to get right and worthy of entire publications.
-Please clarify the prior used for BNN models.
-Please consider using HMC as a baseline for BNN models per task in terms of LLK to compare to DAMS.

Baselines:
- A lot of this paper relies on comparison to baselines, which are chosen to be mostly from the meta-learning field.
However, in practice the goal of the paper is Bayesian Inference in a particular class of models.
Hence:
- please consider adding conditional MNF as a baseline. This would clarify if a simple conditional version of MNF would suffice here compared to the involved scheme proposed in this paper and might show the advantage of DAMS over MNF (effectively the main driver for most of the experiments here).
-Similarly, please consider using conditional NAI flow as a baseline to see how far that gets the reader.


Presentation Suggestions:
-You might want to consider establishing a formal relationship to a hierarchical probabilistic model with plates and show that this is just a way to perform sampling the posterior in a model like: P(y|x, tau) = integral_w P(y|x, w_t) P(w_t|tau) d_w
-This might help the flow of the paper by setting the stage early, as currently I had to read through it halfway to really understand the task before starting from the beginning to absorb the details of the proposed techniques.
-figures only readable in pdf, not in printout. Please enlarge fonts to give 'old school' readers a chance
-page 6  "..could be calculated effectively.." What does effectively mean here? Efficiently?
-page 6 under Theorem 1 typo: 'via the Eulaer scheme' -> Euler scheme

Related Work Suggestions:
- Consider citing "Predictive Uncertainty Quantification with Compound Density Networks" by Kristiadi et al, as it uses a conditional model with multiplicative parametrization successfully. I understand this is per data-point and the meta-learning scenario is focused on the per-dataset setting, but I find them related enough to consider a discussion.
- With regards to inference networks on BNNs, please cite "Latent Projection BNNs: Avoiding weight-space pathologies by learning latent representations of neural network weights" by Pradier et al, which attempts to do this and also discusses related work in more detail than this paper here. It is a hard task to be done well.
-The general form of the ELBO shown here is an instance of "Hierarchical Variational Models" by Ranganath et al, which should also be cited.
-Last but not least a recent paper in an ICML workshop on automatic machine learning  (https://sites.google.com/view/automl2019icml/accepted-papers) had a paper on "Improving Automated Variational Inference with Normalizing Flows" by Webb et al. This method looks a lot like a baseline method for this paper before task adaptation and WGF is considered and would effectively subsume the first batch of experiments entirely. I would propose the authors cite and discuss differences in detail.


Decision:
The paper uses a variety of 'puzzle pieces' that are quite involved on their own right. Putting them together and making it work is nontrivial and the authors demonstrate in their experiments that they get strong performance metrics. However, unfortunately, systematic ablation experiments and detailed analysis for the individual components used here and systematic comparisons to simple baselines are not performed. In addition, the paper is presented as a method for adaptive uncertainty quantification, which as argued above is not demonstrated empirically or else. What the paper does achieve is build a pipeline that gets high predictive performance on a meta-learning setting with lower computational requirements during testing than competing methods. I would suggest the authors focus on that aspect and add the required baselines that would clarify what ablations would do to the system and how the components interact.
As currently presented, I would argue for rejection since I am not sure of the scientific value of the interplay of components here as regarding uncertainty quantification. However, I think this paper is promising for a slightly different story with small experimental adjustments and would encourage the authors to consider that route.


(Edit Post Rebuttal:  revising score given the author response which addressed some of my concerns)

**Experience Assessment:**

I have published in this field for several years.

**Review Assessment: Checking Correctness Of Derivations And Theory:**

I assessed the sensibility of the derivations and theory.

**Review Assessment: Checking Correctness Of Experiments:**

I carefully checked the experiments.

**Review Assessment: Thoroughness In Paper Reading:**

I read the paper thoroughly.

---

> ### Author Response · Authors · 2019-11-15
> **Response to Reviewer1**
>
> Thank you for your insightful comments. We summarize your comments below and try to address them by adding more clarifications and extra experimental results.
>
> Your major concerns:
> 1. The evaluation is focused on #of steps during testing, but not on # of datapoints required for eval on new domains.
> 2. The uncertainty is barely evaluated except in the low-d and toy sine wave illustration.
> 3. Kernels on high-dimensional space.
> 4. Show the performance of meta-sampler with and without sample adapter to clarify the effects of performing sample adaptation versus just using the meta-sampler. And comparing with HMC. The performance of another conditional model? Why choose this conditional model to do sampling?
> 5. Details on the nontrivial ELBO in Sec.3.4.
> 6. Presentation.
> 7. Related work.
>
>
> Our responses:
> 1. We have added additional results showing the performance versus #datapoints during testing. Figure 5 shows the results of evaluations on #datapoints. We show the results of picking 5%, 20% and 30% of training data as new training data and test their accuracies on the same test data, and compare them to NNSGHMC and SGHMC. We also would like to point out that the few shot image-classification task we did in our experiments is exactly designed to demonstrate sample efficiency. Specifically, 1-shot and 5-shot learning use only one and five samples per class to adapt to classification. Our results have well demonstrated the sample efficiency of our method, too.
>
> 2. We have added additional results compared to SGHMC and NNSGHMC. Please see the results in Table 3 and Figure 6, as well as Section D.4 in the Appendix.
>
> 3. We agree that the problem of kernels on high-dimensional space is a long-standing problem. It is exactly this reason that motivates us to reduce the dimension of the parameter space via the multiplicative-parameterization method. This method can reduce the dimensionality significantly. For example, in the smaller LeNet structure we use, the maximum kernel input is only 300 dimensions in our model, compared to more than 240K dimensional parameters in the original space. In our experiments, the dimension of the parameter never exceeds 300, which makes the kernel-based method effective.
>
>      We also want to mention that in the original SVPG (Stein Variational Policy Gradient) paper, the authors even performed kernel computations in the original parameter space of a policy network (100-50-25 hidden units), whose dimension is more than 6000 dimensions, much higher than ours, the authors of SVGD have shown promising results in their cases. Therefore, we believe with our techniques to further reduce parameter dimensionality, the issue would not be a big problem.
>
> 4. We have conducted additional experiments to try to address your comments.
> Figure 4 shows the results of only using meta sampler with and without the sampler adapter. Training from scratch is the result of only using meta sampler. Meta adaptation with DAMS is actually the results of meta sampler with sample adapter.  Also, we add additional experimental results with Bayesian logistic regression,  comparing our DAMS with different structure of the meta generator, including MLP, IAF and NIAF, for BLR. The results clearly demonstrate the effectiveness of our DAMS structure.
>
>
> 5. We agree obtaining and optimizing the ELBO is non-trivial. However, since the ELBO is obtained in a similar way as in the original paper of MNF, we chose not to include details in the text.
>    (1) The prior for BNN is Gaussian prior
>    (2) We have added additional results by replacing DAMS with the original MNF. Figure 4 and Table 2 show the comparisons. In Table 2, it is seen that our method improves over MNF by more than 2% on CIFAR10.
>
> 6. We thank you for your suggestions on the presentation. We will surely revise our draft based on your comments. Due to the need of providing extra experimental results, we did not focus on restructuring the paper for this revision, but surely will consider it in our final version.
>
> 7. Thanks for providing the excellent related works. We have included and discussed them in our revision. (1) "Predictive Uncertainty Quantification with Compound Density Networks"  is an extension of mixture density networks to quantify the predictive uncertainty; (2) "Latent Projection BNNs: Avoiding weight-space pathologies by learning latent representations of neural network weights" which performs inference on a lower dimensional latent space; (3) Hierarchical Variational Models by Ranganath et al has been cited; (4) "Improving Automated Variational Inference with Normalizing Flows"  first samples from a mean-field approximating distribution,  then the samples are transformed by NAF to a more expressive distribution and their approach is different from ours and is hard to scale to very high dimensional problems , e.g., posterior distribution over network.
>
>
> We hope you can check our revision and reconsider your decision. Thank you.

---

> > ### Comment · AnonReviewer1 · 2019-11-15
> > **thank you for the revisions**
> >
> > Thank you for your update, I took note of it and will reflect it in my assessment.

---

> > > ### Author Response · Authors · 2019-11-15
> > > **thanks a lot for your update**
> > >
> > > We thank you for your valuable time and update.

---

### Official Review · AnonReviewer2 · 2019-10-24
**Official Blind Review #2**

**Rating:** 3

**Review:**

This paper proposes a Bayesian meta sampling framework consisting of two main components: a meta sampler and a sample adapter. The meta sampler adopted a NIAF structure to generate meta samples, while the sample adapter adapts the samples based on optimal-transport Bayesian sampling.

What is the advantage of this Bayesian meta learning methods comparing to the popular none Bayesian methods?  It seems the performance of this Bayesian method is inferior to the state-of-the-art meta-learning methods. What are the specific model structures for T and G in NIAF?

The experiments can be improved.  First, the comparison methods didn’t appear in all the results. For example, why PMAML is not compared with in Figure 3, Table 1 and Table 2?
Second, for the standard meta-learning tasks in Table 3, it is better to compare with the state-of-the-art methods.

**Experience Assessment:**

I do not know much about this area.

**Review Assessment: Checking Correctness Of Derivations And Theory:**

I did not assess the derivations or theory.

**Review Assessment: Checking Correctness Of Experiments:**

I assessed the sensibility of the experiments.

**Review Assessment: Thoroughness In Paper Reading:**

I made a quick assessment of this paper.

---

> ### Author Response · Authors · 2019-11-15
> **Response to Reviewer2**
>
> Thanks for your comments. Please see our responses below.
>
> Q: Advantage of this Bayesian meta learning methods.
> A: We have actually stated the advantages of the proposed Bayesian meta sampling over other existing methods in the 3rd paragraph in the Introduction. To emphasize, we argue that most existing methods tackle meta learning from the perspective of “parameter adaptation”, instead of “uncertainty/posterior adaptation” considered in our paper. Without uncertainty adaptation, it could slow down model adaptation or result in inaccurate uncertainty modeling when considering from a Bayesian modeling perspective. Our experimental results indeed show that posterior adaptation could result in better performance compared to existing non-uncertainty adaptation methods.
>
>
> Q: Not the state-of-the-art. No comparisons.
> A: We would like to argue that there have been a lot of meta learning methods, each designing from a different perspective. For example, some belong to gradient-based methods, e.g. MAML and its related variants. However, we note the architecture and pre-trained feature extractor matter a lot in the performance of LEO (Meta-Learning with Latent Embedding Optimization). Other meta learning methods include the graph network based models (e.g., Edge-Labeling Graph Neural Network for Few-shot Learning), the amortized variational inference (e.g., Meta-Learning Probabilistic Inference for Prediction), and the metric-based meta learning methods (e.g., TADAM: Task dependent adaptive metric for improved few-shot learning).
>
> The current state-of-the-art we are aware of is [Finding Task-Relevant Features for Few-Shot Learning by Category Traversal]. However, it is designed from a very different way compared to ours. Specifically, their approach depends on metric-based learning, i.e., support-query similarity, and is restricted to the task of few-shot classification. Our method adopts the most flexible framework of MAML, which can be very general and be applied to different settings such as few-shot learning and reinforcement learning. Therefore, to be fair, we believe one should only compare the most related methods such as MAML, PMAML and ABML as done in our paper. Since our idea is different from the state-of-the-art, we believe we can combine our ideas with the state-of-the-art to achieve even better performance, which is non-trivial thus left for future work.
>
> Q: What are the specific model structures for T and G in NIAF?
> A: The structures for T is MADE, you can refer to (1) MADE: masked autoencoder for distribution estimation, Germain et al.  and (2) Neural Autoregressive Flows (Huang et al.) for more details.
>
> The structure for G is a deep sigmoidal flow (DSF) or deep dense sigmoidal flow (DDSF). Please refer to Neural Autoregressive Flows (Huang et al.) for more details.
>
>
> Q: Why PMAML is not compared with in Figure 3, Table 1 and Table 2?
> A: PMAML is not designed for Bayesian sampling, and Figure 3 is the task of Bayesian sampling for mixture models.
>
> Table 1 is used for evaluating Bayesian logistic regression in a non-meta-sampling setting, thus PMAML is not applicable.
>
> Table 2 is used for evaluating BNN in training and testing tasks, which consists of 5 classes. The training task only consists of one task (which is not applicable for PMAML), which is the setting as “Meta-Learning For Stochastic Gradient MCMC ”.

---

### Author Response · Authors · 2019-11-15
**Major changes in the revision**

We would like to express our appreciation to the reviewers, whose comments help us improve our paper a lot. According to the reviews, we note the major concerns raised are about empirical evaluations. We have done a major revision to address the reviewers’ concerns. We have incorporated our changes into the submission and updated a new version in openreview. The major content added are marked with “red” in this revision for a better comparison. To summarize, below is a list of major changes we have made in the revision by adding more empirical results:

       1. We have compared our DAMS with different network structures for the generator (meta sampler) including MLP, IAF, NIAF on the Bayesian logistic regression task (Table 1).
       2. We have conducted extra experiments to show the sample efficiency of our model, with different percent of training data (Figure 5).
       3. We have compared with SGHMC under different scenarios to validate both sample efficiency and uncertainty (Table 3 and Figure 5 and 6).
       4. We have added a baseline MNF proposed by Louizos and Welling.
       5. We have added more related works and discussions on inference network and multiplicative parametrization.
       6. We have moved the Background section and the experiment of “Meta posterior adaptation” to the Appendix.
       7. We have added additional experiment on uncertainty evaluation via entropy of out-of-sample prediction distributions in the Figure 6 and Appendix.

Our conclusion remains with the additional results. Specific concerns are addressed individually for each reviewer. We believe our extra results have well addressed the concerns raised by the reviewers. We hope the reviewers can check our revision carefully and re-evaluate their decisions.

---

### Decision · Program_Chairs · 2019-12-19

**Decision:**

Accept (Poster)

**Comment:**

This paper presents a meta-learning algorithm that represents uncertainty both at the meta-level and at the task-level. The approach contains an interesting combination of techniques. The reviewers raised concerns about the thoroughness of the experiments, which were resolved in a convincing way in the rebuttal. Concerns about clarity remain, and the authors are *strongly encouraged* to revise the paper throughout to make the presentation more clear and understandable, including to readers who do not have a meta-learning background. See the reviewer's comments for further details on how the organization of the paper and the presentation of the ideas can be improved.